# Remote home cardiotocography: A systematic review and meta-analysis

Jack Le Vance[1,2]*, Adekunle Adeoye[3], Rebecca Man[1,2], Nashwa Eltaweel[2], Leo Gurney[2], R.Katie Morris[1,2], Victoria Hodgetts Morton[1,2]

1 Department of Applied Health Sciences, School of Health Sciences, College of Medicine and Health, University of Birmingham, Birmingham, United Kingdom, 2 Birmingham Women's and Children's NHS Foundation Trust, Birmingham, United Kingdom, 3 The Shrewsbury and Telford Hospital NHS Trust, Shrewsbury, United Kingdom

* j.hamer@bham.ac.uk

## Abstract

Cardiotocography (CTG) is a common investigative modality in obstetrics to evaluate the fetal condition. Advancements in digital technology has enabled the innovation of CTG monitoring for usage in the home setting. This review aims to comprehensively examine the current evidence on the effectiveness and applicability of home antenatal CTG monitoring. MEDLINE, EMBASE, Cochrane, Web of Science, and PubMed databases were searched from inception to June 2025. Primary studies examining home antenatal CTG were included. For randomised controlled trials (RCTs), the joint primary outcomes were perinatal mortality and emergency caesarean section. For observational studies, the feasibility, diagnostic accuracy, qualitative and economic burden of home CTG were evaluated. RCTs were eligible for meta-analysis using risk ratio or mean difference, with 95% confidence intervals. Included observational studies were narratively described due to significant methodological heterogeneity. 39 studies (28 observational, seven RCTs and four qualitative studies), comprising of 7240 participants were included. Home antenatal CTG monitoring was non-inferior to conventional care across all meta-analysed maternal, perinatal and healthcare usage outcomes. GRADE assessments were low/very low quality of evidence. Home CTG monitoring was feasible in several settings and remote interpretation was graded as moderate to excellent. Transmission failures were frequently low but commonly occurred due to infrastructure and/or equipment errors. Remote CTG monitoring demonstrated comparative capabilities to conventional CTG with respect to coincidence and beat-to-beat variability. Overall acceptability ratings were high for patient and providers. Often implementation costs were high but accrued back by non-fixed savings when compared against routine care. High-quality studies were underrepresented, particularly when assessing service-led and safety outcomes. Home antenatal CTG monitoring demonstrates noninferiority to conventional care across several outcomes, representing a promising avenue for antenatal management However,

**Data availability statement:** All data supporting the findings of this study are included in the manuscript and its supplementary information files.

**Funding:** The author(s) received no specific funding for this work.

**Competing interests:** The authors have declared that no competing interests exist.

current evidence is of low quality and additional high-quality evidence with sufficient methodological detail and standardised outcome assessment is required prior to making definitive recommendations.

## Author summary

In an era where the number of pregnant women classified as high-risk is increasing, the demand for regular antenatal fetal monitoring has subsequently surged, which has increased the number of consultations in outpatient maternity services. Developments in digital technology has enabled devices, such as cardiotocography to be used in the home setting, which may alleviate the burden on the outpatient department. However, currently there is limited research which comprehensively examines the available literature on home CTG monitoring. Our review identified 39 papers, comprising of 7240 participants which incorporated a range of different study types and outcomes to be evaluated Meta-analysis of 16 maternal, perinatal and service-led outcomes demonstrated consistent non-inferiority when compared to conventional care across seven randomised controlled trials. However, high risk of bias, poor methodological quality and inconsistent results led to low or very low certainty of evidence using the GRADE approach. Narrative description of observational studies enabled a global assessment of the feasibility, diagnostic accuracy, clinical utility, qualitative and economic burden of home antenatal CTG monitoring. This review provides recommendations to shape future study design, whilst ensuring consistent outcome reporting, ultimately aiming to improve research quality for remote fetal monitoring technology interventions.

## Introduction

Cardiotocography (CTG) is a commonly used investigative modality in maternity care to evaluate fetal condition; it is frequently used in pregnant women classified at higher risk of complications, such as perinatal mortality.. Over recent years, the number of high-risk obstetric patients has substantially increased, primarily attributed to recent guideline changes which have lowered diagnostic thresholds for obstetric conditions, implementation of national initiatives, combined with an ageing maternal population [1–5]. Consequently, this has placed a significant strain on the outpatient service to accommodate the increased demand for serial monitoring, such as antenatal CTG. Concurrently, frequent requirements for hospital monitoring can pose a psychological, emotional and monetary strain for women during the antenatal period [6,7]. Frequent travel to the hospital can also pose an environmental impact which is an importation consideration to mediate following the NHS plan to reach net zero by 2040 [8]. Therefore, discovering pathways to address such concerns is paramount to enhancing the quality of current obstetric provision.

PLOS Digital Health

Advancements in digital medical technology combined with the increased access to wireless internet has enabled the innovation of sophisticated remote CTG technology which can be conducted outside of the hospital setting [9]. The recent pandemic has also had a substantial influence on technology development, as obstetric services now need to consider alternative solutions to current healthcare provision standards [10]. Remote digital maternal-fetal monitoring represents potential advantages by improving patient autonomy and education, whilst potentially reducing hospital attendance and overall economic burden [9,11]. Importantly, remote CTG presents opportunities to improve overall access to fetal monitoring services and aims to address current health outcome disparities seen across the obstetric population [12]. Despite this, widespread implementation has not yet been achieved due to the possible safety risks and cost implications, whilst clearly the appropriate indications for routine usage within pregnancy must be determined. A single prior review on the effectiveness of remote CTG monitoring has demonstrated favourable outcomes, however, combined both remote home and remote in-hospital monitoring at different gestational periods, proving difficulty for researchers to delineate the true benefit of home monitoring [13].

Before home antenatal CTG can be routinely implemented into care practice, it is vital to determine whether it is equivalent or non-inferior to conventional care primarily with respect to clinical outcomes, such as the identification of fetal compromise. This enables clinicians to understand the current evidence and elucidate whether this monitoring strategy is beneficial and deemed worth to continue to research. Outcomes such as perinatal mortality is a pertinent outcome to assess in novel monitoring strategies and can frequently be a definitive reason to discontinue an intervention, however, it can also be troublesome to comprehensively evaluate due to its rare occurrence. Therefore, in order to adequately determine the entire effectiveness of obstetric monitoring strategies, such as home CTG, it is essential to additionally employ a global stance and examine all relevant clinical outcomes, including its impact on healthcare usage. This further encompasses the feasibility, acceptability and economic implications associated with remote CTG. In the case of home CTG, a superior participant experience and beneficial impact on service led outcomes, combined with a reduction in associated costs may be required to consider this intervention as an acceptable alternative to routine care., To the best of our knowledge, there are currently no systematic reviews collating multiple outcome data regarding home CTG monitoring. Therefore, we aimed to comprehensively examine such outcomes to formally evaluate the applicability of home antenatal CTG monitoring for the obstetric population.

## Methods

### Study registration

This systematic review adhered to the Preferred Reporting Items for Systematic Reviews and Meta-Analysis (PRISMA) guidelines (S1 Table) [14]. A protocol for this review was registered with PROSPERO (CRD42024591303).

### Inclusion and exclusion criteria

Inclusion and exclusion criteria can be visualised within Table 1. Principally, all primary research studies on remote antenatal CTG usage within obstetrics, whereby clinical assessment was undertaken in the home setting with additional tele-transmission were included. Studies using alternative terminology aside from CTG, such as non-stress test, but the principle of fetal monitoring remained the same, were included. Remote intrapartum CTG studies were excluded. Studies using alternative electric monitoring devices, such as fetal electrocardiogram, phonocardiotocography and magnetocardiography, whereby the data was not converted to a traditional, interpretable CTG recording, were excluded. Conference abstracts, communication papers, editorials, systematic reviews, narrative reviews and single case reports were excluded.

### Search strategy, study selection and data extraction

MEDLINE, EMBASE, the Cochrane Database of Clinical Trials (CENTRAL), Web of Science, and PubMed databases were searched from inception to November 2024. An updated search was later conducted from November 2024 to 30th

**Table 1. Inclusion and exclusion criteria.**

|  | Inclusion | Exclusion |
|---|---|---|
| **Publication type** | • Full text articles<br>  ◦ Any of: randomised controlled trial, feasibility study, prospective/retrospective observational study, validation study, pilot study, diagnostic accuracy study, case series (two or more patients), qualitative study, economic evaluation study. | • Any of: conference abstract, communication paper, letters to the editor/editorials, systematic review, narrative review, single case reports. |
| **Participants** | • High or low risk pregnant women in the antenatal period. | • Any of: pregnant women in the intrapartum period, simulated patients, non-pregnant women, neonates, infants. |
| **Setting** | • Home setting | • Hospital or pre-hospital setting. |
| **Intervention** | • Any indication for an obstetric antenatal CTG with transmission of data sent to reviewer for review. | • Remote electronic devices which do not use CTG, or does not convert the fetal heartrate data into a recording which is identical to CTG trace. |
| **Comparator** | • Not required to have a comparator for inclusion. However, if a study has a comparator then to be compared against a reference standard such as routinely impleVmented antenatal CTG care. | |
| **Outcomes** | • Outcomes relating to the provision of antenatal obstetric CTG. Any of: clinical outcomes, economic outcomes, qualitative outcome. | • Outcomes that are not relevant to obstetric CTG. |

June 2025. Search terms and functions were appropriately adapted for each database with medical subject headings (MeSH) and text words used. The research strategy for MEDLINE and Embase can be seen in S2 Table. Reference lists were additionally screened to identify additional grey literature citations eligible for inclusion. No language restrictions were applied. Covidence software (Veritas Health Innovation M, Australia) was used to manage title, abstract, and full manuscript screening. This process was repeated for subsequent data extraction. All titles, abstracts and full texts were reviewed independently by two authors (JLV, AA, RM, NE) and conflicts were discussed with an additional independent author (LG). For eligible studies, data was extracted into a pre-designed spreadsheet. Data extracted from each study included the; author, year of publication, study location, patient demographics, sample size, indication for remote CTG usage, gestational age of usage, technology used, technical details (transmission delay, failure and signal loss), and additional study specific outcomes. Data extraction was individually performed by two independent researchers (JLV, AA, RM, NE) Any discrepancies were resolved by consensus between researchers.

## Risk of bias assessment

For observational studies assessing remote CTG devices, the methodological quality of these studies was assessed using an adapted form of the QUADAS-2 checklist from NICE and Marsh-Feiley et al. (S3 Table) [15,16]. Seven domains, adapted for obstetric studies, were assessed to determine bias: 1) Generic quality standards; 2) Participant selection bias; 3) Index test; 4) Reference standard; 5) Flow and timing; 6) Telemedicine/technology specific items; 7) Overall applicability. Risk of bias for qualitative studies was performed using the most relevant and recent critical appraisal checklists from the Joanna Briggs Institute [17]. The Cochrane risk of bias tool (RoB 2) was used to assess bias in randomised controlled trials [18]. Individual study quality was assessed using the pre-defined criteria to justify each study as either high-risk, low-risk or with some concerns. Risk of bias for each study was individually assessed by two reviewers (JLV, AA, RM, NE). Any discrepancies with bias assessment were arbitrated with an additional independent author (LG).

## Outcomes

Outcomes assessed were stratified based on the type of included study. For randomised controlled trials, the joint primary maternal and perinatal outcomes were perinatal mortality, defined as fetal and neonatal mortality, and emergency

caesarean section, as these outcomes were deemed most clinically pertinent when assessing remote antenatal monitoring strategies. Secondary outcomes were evaluated on an a priori approach, taking into account the availability of data for clinically important outcomes. This included mode of delivery, low Apgar score (<7), gestational age at delivery, preterm birth, birthweight and neonatal intensive care (NICU) admission. Additionally, outcomes relating to healthcare usage were assessed. For observational studies, outcomes related to feasibility, diagnostic accuracy, clinical utility, qualitative and economic burden were examined, with additional input from included trials. Feasibility outcomes included, transmission delays, transmission failures, the reported quality of remote monitoring and frequency of interpretable monitoring. The definition of interpretable monitoring was at the discretion of each included study. Clinical utility outcomes included examining the impact of remote CTG on outpatient attendances, outpatient clinic time and hospital admission.

### Certainty of evidence

The GRADE approach was used to assess the certainty of the body of evidence for each outcome in the randomised controlled trials only, as this was deemed most appropriate as the effectiveness of the intervention was being considered [19,20].

### Data analysis

Inclusion of two or more randomised controlled trials with identical clinical outcomes were eligible for meta-analysis using Review Manager 5.4 (RevMan) [21]. Dichotomous outcome data was assessed with 2 × 2 tables constructed to calculate risk ratios (RR) and 95% confidence intervals (CI). Mantel–Haenszel random fixed-effects models was utilised for meta-analysis given the methodological heterogeneity with regards to device type, participant population and antenatal care pathways provided. Zero-cell adjustments to 0.5 were be used if required. Continuous outcome data were compared using mean difference (MD) and 95% CI, using the inverse variance fixed-effects model. Continuous variables reported in studies as median and range were translated to proxy mean and standard deviation (SD) using methodological equations described by Hozo et al. and Wan et al [22,23]. Calculated means and SD for relevant groups from each study were combined together for an overall mean and SD using the Cochrane method for combing means [24]. Summary RR and MD were configured with forest plots, allowing visual inspection and quantitative assessment for heterogeneity using $I^2$. $I^2$ values below 25% are considered low, approximately 50% considered moderate, and above 75% considered high levels of heterogeneity [25]. When moderate and high heterogeneity was noted, individual study methodology was examined to determine potential reasons for why this occurred. Subgroup analyses were intended for meta-analysed outcomes where possible, including remote monitoring frequency, type of person completing the remote monitoring, risk of bias judgement and age of study. Due to the small number of included studies (fewer than 10), publication bias was not examined, as test power is frequently too low to distinguish chance from true asymmetry on funnel plot assessment [26].

For additional included observational studies, exploring a wide range of outcomes, meta-analysis of the available data was not possible, and a descriptive synthesis of data is therefore provided. This is primarily due to high risk of bias of included studies and significant heterogeneity in study methodology and available outcome measures. Consolidated evidence was not graded but rather assessed quantively throughout this paper.

## Results

The database and grey literature searches identified 7695 abstracts. A total of 3352 duplicate abstracts were removed, leaving 4343 abstracts for screening. A total of 175 full texts were reviewed, from which 39 papers (28 clinical observational studies, seven randomised controlled trials and four qualitative studies), comprising 7240 participants, recruiting across 13 countries were eligible for inclusion (Fig 1) [9,11,27–63]. The most common reason for exclusion during full text examination was that the usage of CTG was not conducted in the home setting. Excluded studies following full-text screening are presented in S4 Table.

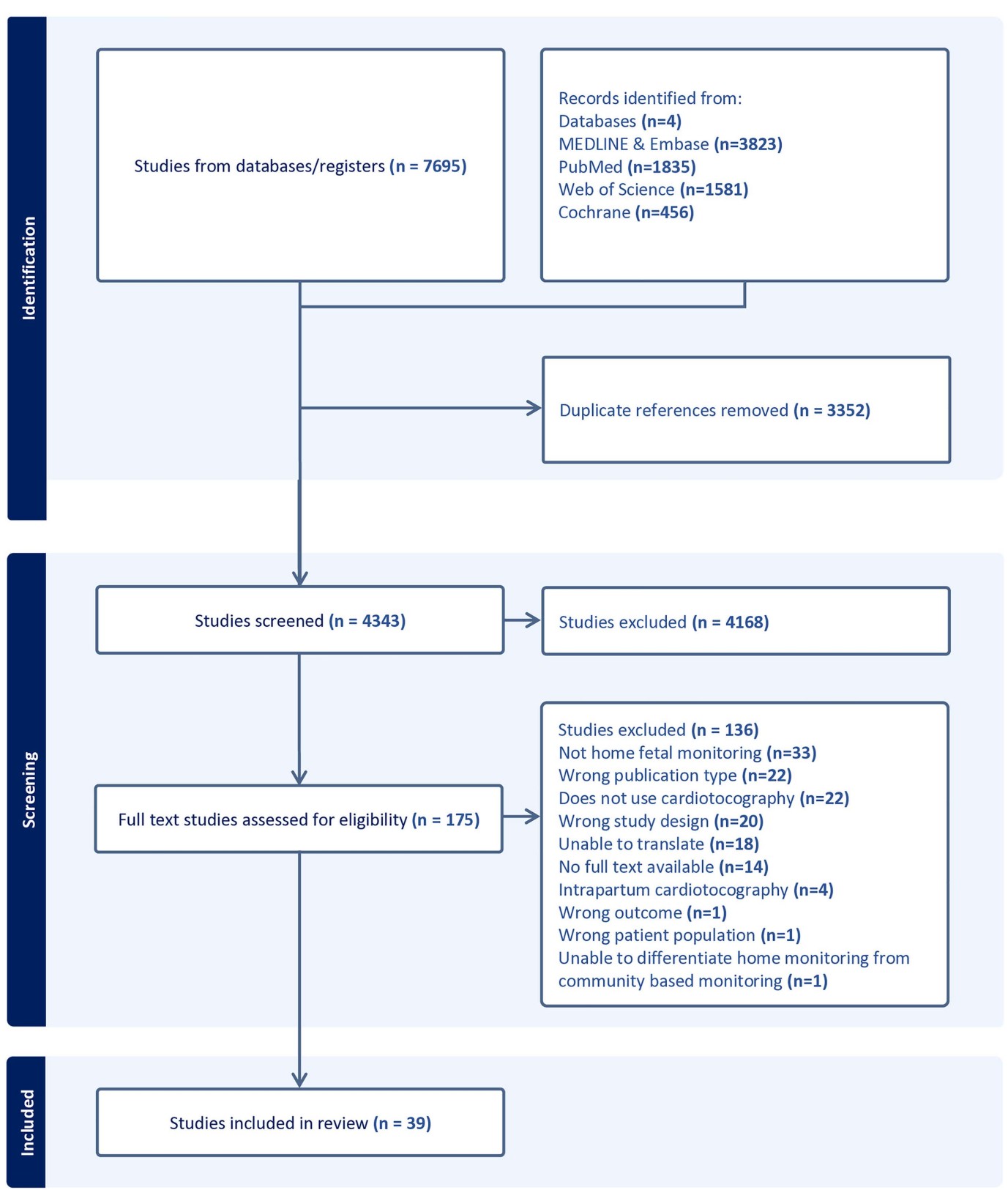

**Fig 1. PRISMA flow diagram.**

All articles were published between 1986 and 2025, with 46% of publications occurring between 2019–2025. There were no included studies from low- and middle-income countries. One study occurred in each of: Canada, Italy, Norway and Poland. Two studies in each of: France, Denmark and Isreal. Three studies in Australia. Four studies in each of: China and the United States of America. Five studies in Japan, six in the Netherlands and seven in the United Kingdom (UK). A graphical dispersion of study recruitment can be seen in Fig 2.

Included studies were highly heterogenous in design, methodology and the outcomes measured. Study sample sizes ranged from 3 to 2860 participants; however, 74% of studies included 100 participants or fewer. Remote monitoring predominantly occurred within the third trimester (n = 21), with seven studies further assessing in the late second trimester and 11 studies not disclosing the gestation of monitoring. Participants complicated with high-risk obstetric conditions were primarily evaluated in the included studies (n = 21), with six studies including both low and high-risk participants. Five studies only included low-risk participants, and seven studies provided no such information.

Observational studies were broadly categorised into three main subtypes. Firstly, feasibility studies considered the capabilities and specifications of remote CTG monitoring to transmit and receive information that was deemed clinically useful [28,32,33,36,37,39–42,44–46,48–50,54–58,61–63]. Secondly, diagnostic accuracy studies evaluated the diagnostic capabilities of remote CTG monitoring, frequently in relation to conventional CTG monitoring [52,53]. Finally, pilot studies were those conducted in a clinical setting with additional defined control groups, without randomisation [38,51].

Completion of remote CTG monitoring in the home setting was frequently undertaken by the participant, however, in a subset of included studies, the community midwife completed the monitoring [30,31,34–36,47,48,58]. Despite device education playing a fundamental role in participant competency for remote monitoring completion, only 39% of studies detailed any information of device training [9,11,28,33,39,45,46,52–54,56,57,62,63]. Review of remote CTG monitoring data was either conducted in a synchronous [9,33,37,42,50,51,53,56,57,62,63] or asynchronous format [27,33,35,39,43,45,47,48,53]. However, frequently this was not clearly reported by studies [11,27,29–31,35,37,38,41,43,45,47,52,55,59–61]. Transmission of remote monitoring data was either by internet or telephone lines, with the majority of older included studies, transmitting with the latter modality. A detailed description of the study characteristics can be found in S5 Table.

## Randomised controlled trials and GRADE assessment

Seven randomised controlled trials comparing home CTG with conventional care were included within this review, enabling pooled meta-analysis on 16 maternal, perinatal and healthcare outcomes [9,30,34,35,47,59,60]. Frequency of remote monitoring varied from up to four times per day [58], to daily [9,30,47], to twice weekly [60], or variable frequency based on the participants antenatal condition and needs [34,35]. When documented, conventional care comprised of either outpatient CTG monitoring [58,59] or inpatient admission [9,30,35,47]. Home CTG monitoring was either performed by participants [9,60], midwives [34,35], or both [30,47]. Birnie et al. and Monincx et al. were separate publications but noted to contain the same study data. Both studies were included in this review as particular outcome data was available exclusively in each paper. However, both studies were not included in the same meta-analysis in order to avoid duplication of data. The summary of findings including the GRADE assessment is presented in Table 2.

Overall, the certainty of evidence by GRADE assessment was either low or very low for all meta-analysed outcomes. Several studies were of high risk of bias using the Cochrane RoB 2 tool. Despite all but one study assessed remote monitoring in high-risk participants, there were variations in study design and methodology as noted above, which influenced the overall directness and subsequent impact on grading. Inconsistency and imprecision were moderate in several outcomes with variable point estimates and wide confidence intervals noted. Furthermore, heterogeneity was substantially high in a subset of outcomes, however with few studies included the validity of the presented heterogeneity is limited.

Regarding the primary maternal outcome, calculation of the absolute risk for emergency caesarean section demonstrated an increase of 21 events per 1000 women for those randomised to remote monitoring versus conventional care.

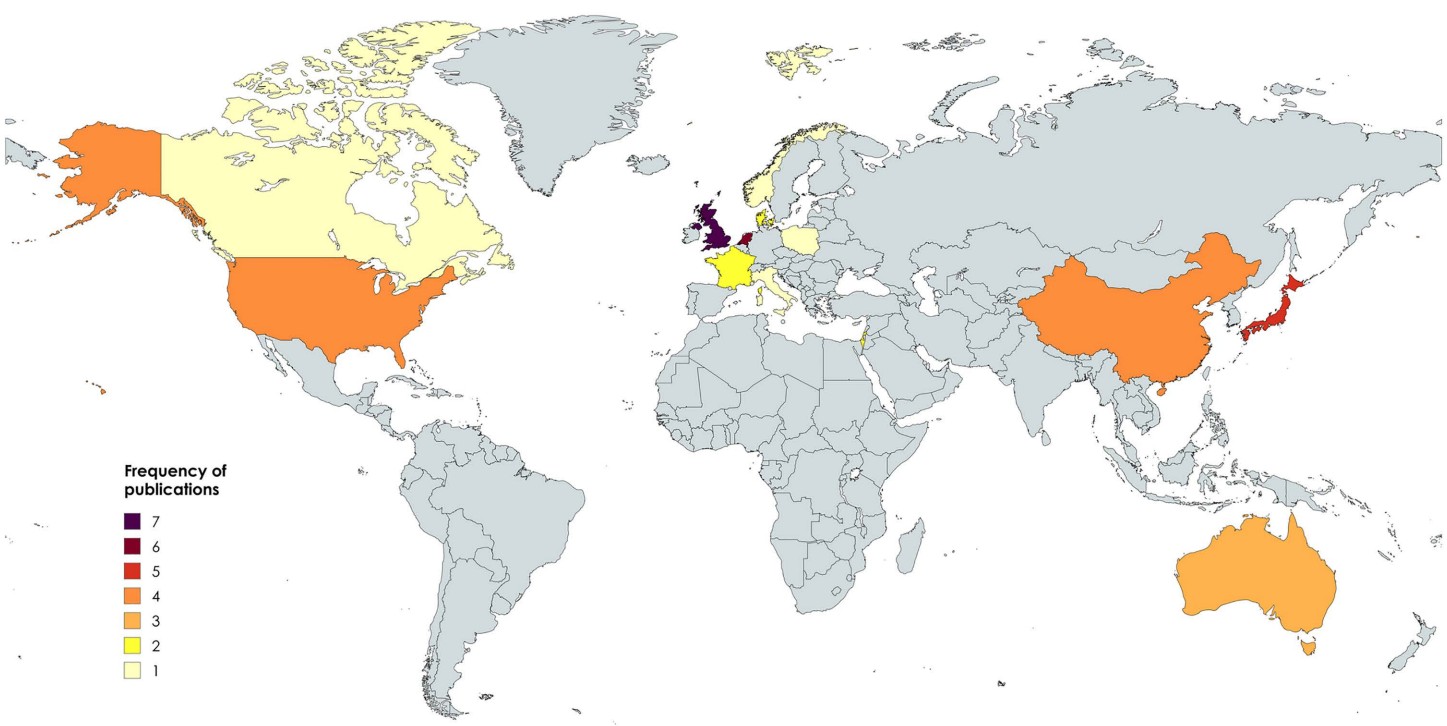

**Fig 2. World map graphical representation of included studies recruitment origin.** Produced at https://www.mapchart.net/world.html.

However, results were non-significant (RR 1.08 [95% CI 0.67 – 1.73], $I^2=0\%$). Furthermore, there were no significant differences in the rate of overall caesarean section, elective caesarean section, induction of labour, spontaneous delivery or instrumental delivery between each group. Statistical heterogeneity was low within all groups (Fig 3). Two studies additionally assessed patient wellbeing using the Edinburgh Postnatal Depression Scale (EPDS) [9,60]. Meta-analysis of two studies demonstrated no statistical significance, with high heterogeneity was noted (MD -1.55 [95% CI 3.90-0.80]; $I^2=94\%$) (Fig 4). However, with few studies included in this analysis, the validity of the presented heterogeneity is limited. Bekker et al. measured the EDPS score at four weeks postnatal whereas Zhou et al. measured at one week postnatal [9,60].Furthermore, the control group in Bekker et al. consisted of inpatient admission compared to outpatient assessment in Zhou et al.; both potential reasons for the high heterogeneity seen.

Regarding the primary perinatal outcome, calculation of the absolute risk for perinatal mortality demonstrated an increase of 11 events per 1000 women for those randomised to remote monitoring versus conventional care. However, results were non-significant (RR 1.97 [95% CI 0.37 – 10.58], $I^2=0\%$). Results were moderately affected by a single study, which on closer inspection included slightly more participants with fetal anomaly and those complicated with preterm rupture of membranes which were randomised to the remote monitoring group [9]. This may attribute to the slight increase in cases of perinatal mortality seen. Additionally, there were no significant differences in the rates of preterm birth, low Apgar scores at 1 and 5 minutes, neonatal admission, gestational age at delivery and birthweight between groups (Figs 5–7). Statistical heterogeneity was low within all groups except preterm birth which was moderate ($I^2=66\%$). Bekker et al. may have had more preterm deliveries in the control group as the participants were assigned to inpatient admission which likely inadvertently led to a greater change of earlier intervention [9]. However, from the two included studies only Zhou et al. defined preterm birth, defining as delivery between 28–37 gestational weeks [60].

**Table 2. Summary of findings table. Effect estimates and GRADE assessment of randomised controlled trials examining maternal and perinatal outcomes for remote CTG monitoring versus to standard care.**

| Outcomes | Total participants (number of studies) | Absolute risk (95% CI) | | | Relative effect (95% CI) | P Value | $I^2$ | Certainty of evidence (GRADE) |
|---|---|---|---|---|---|---|---|---|
| | | Remote group | Control | Difference | | | | |
| **Maternal outcomes** | | | | | | | | |
| Overall caesarean section | 1088 (4) | 379 per 1000 (0.339 – 0.419) | 346 per 1000 (0.305 – 0.386) | 0.03 (-0.02 – 0.09) | RR 1.11 (0.95 – 1.29) | 0.20 | 0% | ⊕⊕○○ LOW[a] |
| Emergency caesarean section | 407 (3) | 157 per 1000 (0.109 – 0.206) | 136 per 1000 (0.087 – 0.185) | 0.01 (-0.05 -0.08) | RR 1.08 (0.67 – 1.73) | 0.76 | 0% | ⊕⊕○○ LOW[b] |
| Elective caesarean section | 207 (2) | 60 per 1000 (0.017 – 0.104) | 110 per 1000 (0.046 – 0.174) | -0.05 (-0.13 – 0.03) | RR 0.56 (0.22 – 1.42) | 0.22 | 0% | ⊕⊕○○ LOW[c] |
| Induction of labour | 288 (3) | 302 per 1000 (0.231 – 0.373) | 372 per 1000 (0.289 – 0.456) | -0.05 (-0.17 – 0.08) | RR 0.84 (0.60 – 1.17) | 0.29 | 0% | ⊕⊕○○ LOW[d] |
| Spontaneous delivery | 231 (2) | 740 per 1000 (0.661 – 0.818) | 741 per 1000 (0.660 – 0.822) | 0.00 (-0.12 – 0.11) | RR 1.00 (0.86 – 1.17) | 0.99 | 0% | ⊕⊕○○ LOW[e] |
| Instrumental delivery | 231 (2) | 76 per 1000 (0.028 – 0.123) | 63 per 1000 (0.018 – 0.107) | 0.02 (-0.04 – 0.08) | RR 1.22 (0.47 – 3.18) | 0.68 | 0% | ⊕⊕○○ LOW[f] |
| EPDS total score | 1000 (2) | Overall mean (SD): 7.34 (5.62) | Overall mean (SD): 9.56 (5.02) | | MD -1.55 (-3.90 – 0.80) | 0.19 | 94% | ⊕○○○ VERY LOW[g] |
| **Perinatal outcomes** | | | | | | | | |
| Perinatal mortality | 351 (2) | 23 per 1000 (0.001 – 0.044) | 12 per 1000 (0.000 – 0.027) | 0.01 (-0.02 – 0.04) | RR 1.97 (0.37 – 10.58) | 0.43 | 0% | ⊕⊕○○ LOW[h] |
| Preterm birth | 1000 (2) | 224 per 1000 (0.187 – 0.261) | 222 per 1000 (0.186 – 0.258) | -0.02 (-0.13 – 0.10) | RR 1.00 (0.70 – 1.43) | 0.99 | 66% | ⊕⊕○○ LOW[i] |
| Apgar score <7 at 1 minute | 1032 (3) | 162 per 1000 (0.130 – 0.193) | 182 per 1000 (0.148 – 0.215) | -0.02 (-0.07 – 0.02) | RR 0.89 (0.68 – 1.16) | 0.39 | 0% | ⊕⊕○○ LOW[j] |
| Apgar score <7 at 5 minutes | 1392 (5) | 106 per 1000 (0.083 – 0.128) | 127 per 1000 (0.102 – 0.152) | -0.02 (-0.06 – 0.03) | RR 0.84 (0.63 – 1.12) | 0.23 | 26% | ⊕⊕○○ LOW[k] |
| NICU admission | 351 (2) | 203 per 1000 (0.144 – 0.263) | 247 per 1000 (0.183 – 0.311) | -0.04 (-0.13 – 0.05) | RR 0.82 (0.56 – 1.21) | 0.32 | 0% | ⊕⊕○○ LOW[l] |
| Gestational age at delivery (weeks) | 1288 (5) | Overall mean (SD): 38.62 (1.88) | Overall mean (SD): 38.73 (1.81) | | MD -0.15 (-0.56 – 0.27) | 0.48 | 0% | ⊕⊕○○ LOW[m] |
| Birthweight (grams) | 1081 (3) | Overall mean (SD): 3190.61 (530.94) | Overall mean (SD): 3159.60 (563.31) | | MD 113.69 (-48.90 –276.27) | 0.17 | 0% | ⊕⊕○○ LOW[n] |
| **Healthcare usage** | | | | | | | | |
| Inpatient admissions | 138 (2) | Overall mean (SD): 0.84 (1.03) | Overall mean (SD): 1.27 (0.97) | | MD -0.52 (-1.38 – 0.34) | 0.24 | 85% | ⊕⊕○○ LOW[o] |
| Length of inpatient admission (days) | 231 (2) | Overall mean (SD): 1.72 (2.92) | Overall mean (SD): 6.56 (6.72) | | MD -4.17 (-9.15 – 0.80) | 0.10 | 92% | ⊕○○○ VERY LOW[p] |

**GRADE framework for rating the strength of evidence**

- **High certainty**: available evidence provides a high level of confidence, and it is unlikely that further research will significantly change the confidence in the estimate.
- **Moderate certainty**: available evidence is sufficient to support a conclusion, but further research may still impact the confidence in the estimate.
- **Low certainty**: available evidence is limited, and additional research is likely to have an important impact on the confidence in the estimate.
- **Very low certainty**: available evidence is insufficient to support any firm conclusions, and future research is expected to have a significant impact on the confidence in the estimate.

**Explanations**

- [a]The proportion of information from studies at high risk of bias is sufficient to affect the interpretation of results. Inconsistency was moderate as pooled results largely attributed by a single study, whereby the point estimate was dissimilar to the other three included studies. However, confidence intervals overlapping and $I^2$ was 0% demonstrating significant heterogeneity is likely absent. Variation in monitoring frequency, personnel completing remote monitoring and pathways of care in the control groups is noted. Publication bias could not be formally assessed by funnel plot assessment.
- [b]High risk of bias was graded in one included study. Moderate inconsistency noted with slight variation in point estimates, however, confidence intervals overlapping and $I^2$ was 0% demonstrating minimal heterogeneity. Differences in monitoring frequency and personnel completing the remote monitoring were noted. Publication bias could not be formally assessed by funnel plot assessment.

*(Continued)*

- [c]One out of the two included studies was rated high risk of bias. Point estimates suggesting effects in the same direction with overlapping confidence intervals, suggesting low inconsistency. However, confidence intervals are noted to be wide, limiting the certainty of evidence. $I^2$ was 0% demonstrating significant heterogeneity is likely absent. Populations completing remote monitoring and pathways of the control group were similar in included studies, however, frequency of monitoring varied significantly. Publication bias could not be formally assessed by funnel plot assessment.

- [d]High risk of bias graded in one of the included studies. Inconsistency was moderate with slight variation in point estimates; however, confidence intervals overlap and $I^2$ was 0% representing significant heterogeneity was likely absent. Variation in the monitoring frequency, personnel completing remote monitoring and pathways of care in the control groups varied. Publication bias could not be formally assessed by funnel plot assessment.

- [e]Neither included study was graded as a high risk of bias. Slight variation in point estimates was noted, representing moderate inconsistency. However, confidence intervals are narrow and overlapping. $I^2$ was 0% indicating minimal heterogeneity. There remain differences in frequency of monitoring, personnel completing remote monitoring and standards of care in the control groups. Publication bias could not be formally assessed by funnel plot assessment.

- [f]Neither included study was graded as a high risk of bias. Moderate inconsistency noted with wide variation in point estimates and pooled results significantly attributed by a single study. However, confidence intervals do overlap and $I^2$ was 0%. There remain differences in frequency of monitoring, personnel completing remote monitoring and standards of care in the control groups. Publication bias could not be formally assessed by funnel plot assessment.

- [g]The proportion of information from studies at high risk of bias is sufficient to affect the interpretation of results. Inconsistency was high with non-overlapping point estimates and confidence intervals. Overall, the pooled confidence interval was wide and $I^2$ was 94% indicating substantial heterogeneity. Frequency of remote monitoring and standard of care for the control groups varied substantially. Publication bias could not be formally assessed by funnel plot assessment.

- [h]Neither included study was graded as a high risk of bias. Inconsistency was moderate as there was slight variation in the point estimates seen, however, confidence intervals overlapped and $I^2$ was 0%, indicating minimal heterogeneity. However, confidence intervals are noted to be wide, due to the small event numbers in the intervention and control groups. Frequency of monitoring and standard of care in the control groups were similar between studies, however, personnel undertaking the remote monitoring episodes varied. Publication bias could not be formally assessed by funnel plot assessment.

- [i]High risk of bias was graded in one included study. Variation of point estimates noted, however, confidence intervals were narrow and overlapping. $I^2$ was 66% indicating moderate heterogeneity. Type of personnel completing the remote monitoring episodes was similar, however, frequency of monitoring and pathways for control groups varied substantially. Publication bias could not be formally assessed by funnel plot assessment.

- [j] High risk of bias was graded in one included study. Variation of point estimates noted and pooled results largely attributed by a single study. Confidence intervals overlapping and $I^2$ was 0% indicating significant heterogeneity is likely absent. Variation in monitoring frequency, personnel completing remote monitoring and pathways of care in the control groups is noted. Publication bias could not be formally assessed by funnel plot assessment.

- [k]The proportion of information from studies at high risk of bias is sufficient to affect the interpretation of results. Moderate inconsistency noted with variation in point estimates. Pooled results largest attributed by a single study. Confidence intervals overlapped, however, are wide in one included study. $I^2$ was 26% indicating low heterogeneity. Variation in the clinical care provided to the control groups, frequency of monitoring and type of personnel completing the remote monitoring episodes was noted. Publication bias could not be formally assessed by funnel plot assessment.

- [l]Neither included study was graded as a high risk of bias. Inconsistency was low, with minimal variation in point estimates seen and overlapping narrow confidence intervals. $I^2$ was 0% demonstrating significant heterogeneity is likely absent. Frequency of monitoring and standard of care in the control groups were similar between studies, however, personnel undertaking the remote monitoring episodes varied. Publication bias could not be formally assessed by funnel plot assessment.

- [m]The proportion of information from studies at high risk of bias is sufficient to affect the interpretation of results. Variation in point estimates can be seen indicating inconsistency. C

- However, confidence intervals overlapping and $I^2$ was 0% demonstrating absent heterogeneity. Variation in monitoring frequency, personnel completing remote monitoring and pathways of care in the control groups is noted. Publication bias could not be formally assessed by funnel plot assessment.

- [n]High risk of bias was seen in two of the three included studies. Point estimates demonstrated minimal variation, with overlapping confidence intervals. $I^2$ was 0% indicating significant heterogeneity is likely absent. Frequency of monitoring and pathway of those randomised to the control group varied substantially. Publication bias could not be formally assessed by funnel plot assessment.

- [o]One included study graded as high risk of bias. Point estimates demonstrating slight variation and confidence intervals overlapping. However, I2 was 85% indicating high heterogeneity. Frequency of remote monitoring and type of personnel completing the remote monitoring were similar between studies. Publication bias could not be formally assessed by funnel plot assessment.

- [p]The proportion of information from studies at high risk of bias is sufficient to affect the interpretation of results. Point estimates significantly whereby confidence intervals did not overlap and $I^2$ was 92% indicating substantial heterogeneity. Significant variations in standard of care in the control group and frequency of monitoring seen. Publication bias could not be formally assessed by funnel plot assessment.

  CI, confidence interval; EPDS, Edinburgh Postnatal Depression Scale; GRADE: GRADE Working Group grades of evidence (see explanations) MD, mean difference; n, number; NICU, neonatal intensive care unit; RR, risk ratio; SD, standard deviation.

*A negative relative effect favours remote monitoring whereas a positive relative effect favours the control group.

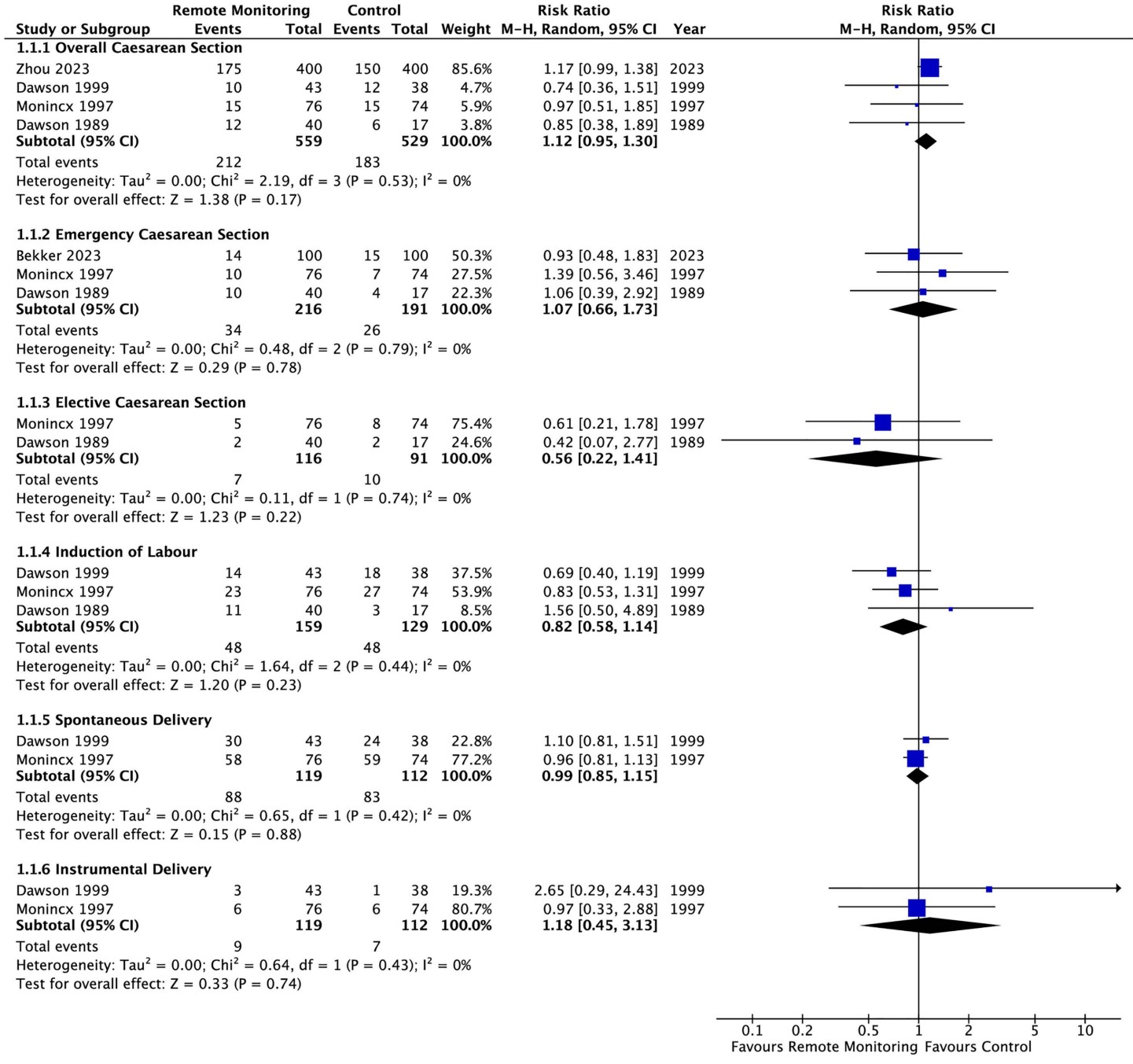

**Fig 3. Summary risk ratios presented as forest plots for assessed maternal outcomes.**

Outcomes relating to healthcare usage were assessed in four of the included randomised controlled trials [30,34,35,47]. However, only two outcomes were available for meta-analysis. There was no statistically significant difference in the number of inpatient admissions per participant during the study period and the length of inpatient stay were seen in those randomised to remote monitoring versus the control group (MD -0.52 [95% CI -1.38 – 0.34] and MD -4.17 [95% CI -9.15 – 0.80] respectively) (Fig 8). However, heterogeneity was high for both outcomes ($I^2 = 85\%$ and $I^2 = 92\%$ respectively).

PLOS Digital Health

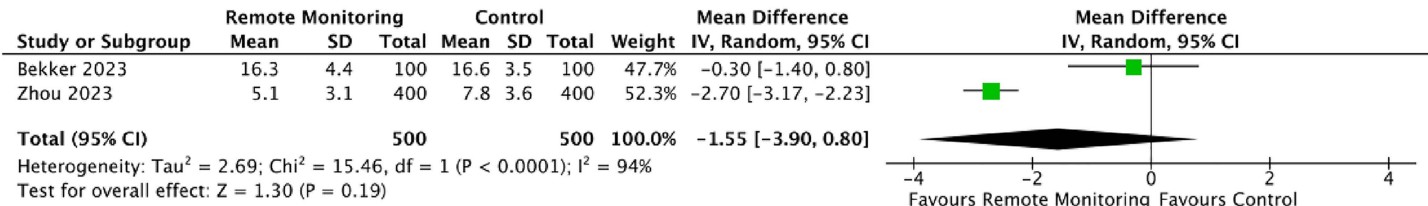

| Study or Subgroup | Remote Monitoring Mean | SD | Total | Control Mean | SD | Total | Weight | Mean Difference IV, Random, 95% CI |
|---|---|---|---|---|---|---|---|---|
| Bekker 2023 | 16.3 | 4.4 | 100 | 16.6 | 3.5 | 100 | 47.7% | −0.30 [−1.40, 0.80] |
| Zhou 2023 | 5.1 | 3.1 | 400 | 7.8 | 3.6 | 400 | 52.3% | −2.70 [−3.17, −2.23] |
| Total (95% CI) | | | 500 | | | 500 | 100.0% | −1.55 [−3.90, 0.80] |

Heterogeneity: Tau² = 2.69; Chi² = 15.46, df = 1 (P < 0.0001); I² = 94%
Test for overall effect: Z = 1.30 (P = 0.19)

**Fig 4. Summary mean difference presented as forest plots for the assessment of patient wellbeing using the Edinburgh Postnatal Depression Scale (EPDS).**

| Study or Subgroup | Remote Monitoring Events | Total | Control Events | Total | Weight | Risk Ratio M–H, Random, 95% CI | Year |
|---|---|---|---|---|---|---|---|
| **2.1.1 Perinatal Mortality** | | | | | | | |
| Bekker 2023 | 3 | 100 | 1 | 100 | 60.0% | 3.00 [0.32, 28.35] | 2023 |
| Monincx 1997 | 1 | 77 | 1 | 74 | 40.0% | 0.96 [0.06, 15.08] | 1997 |
| Subtotal (95% CI) | | 177 | | 174 | 100.0% | 1.90 [0.33, 10.85] | |
| Total events | 4 | | 2 | | | | |

Heterogeneity: Tau² = 0.00; Chi² = 0.40, df = 1 (P = 0.53); I² = 0%
Test for overall effect: Z = 0.72 (P = 0.47)

| | | | | | | | |
|---|---|---|---|---|---|---|---|
| **2.1.2 Preterm Birth** | | | | | | | |
| Bekker 2023 | 56 | 100 | 65 | 100 | 57.7% | 0.86 [0.69, 1.08] | 2023 |
| Zhou 2023 | 56 | 400 | 46 | 400 | 42.3% | 1.22 [0.85, 1.75] | 2023 |
| Subtotal (95% CI) | | 500 | | 500 | 100.0% | 1.00 [0.70, 1.43] | |
| Total events | 112 | | 111 | | | | |

Heterogeneity: Tau² = 0.05; Chi² = 2.90, df = 1 (P = 0.09); I² = 66%
Test for overall effect: Z = 0.01 (P = 0.99)

| | | | | | | | |
|---|---|---|---|---|---|---|---|
| **2.1.3 Apgar Score <7 at 1 minute** | | | | | | | |
| Zhou 2023 | 60 | 400 | 72 | 400 | 73.9% | 0.83 [0.61, 1.14] | 2023 |
| Dawson 1999 | 6 | 43 | 7 | 38 | 7.3% | 0.76 [0.28, 2.06] | 1999 |
| Monincx 1997 | 18 | 77 | 14 | 74 | 18.8% | 1.24 [0.66, 2.30] | 1997 |
| Subtotal (95% CI) | | 520 | | 512 | 100.0% | 0.89 [0.68, 1.17] | |
| Total events | 84 | | 93 | | | | |

Heterogeneity: Tau² = 0.00; Chi² = 1.34, df = 2 (P = 0.51); I² = 0%
Test for overall effect: Z = 0.84 (P = 0.40)

| | | | | | | | |
|---|---|---|---|---|---|---|---|
| **2.1.4 Apgar Score <7 at 5 minutes** | | | | | | | |
| Zhou 2023 | 52 | 400 | 53 | 400 | 49.3% | 0.98 [0.69, 1.40] | 2023 |
| Bekker 2023 | 8 | 100 | 8 | 100 | 16.3% | 1.00 [0.39, 2.56] | 2023 |
| Wang 2019 | 10 | 80 | 24 | 80 | 26.5% | 0.42 [0.21, 0.81] | 2019 |
| Dawson 1999 | 1 | 43 | 1 | 38 | 2.4% | 0.88 [0.06, 13.65] | 1999 |
| Monincx 1997 | 3 | 77 | 2 | 74 | 5.5% | 1.44 [0.25, 8.38] | 1997 |
| Subtotal (95% CI) | | 700 | | 692 | 100.0% | 0.80 [0.52, 1.23] | |
| Total events | 74 | | 88 | | | | |

Heterogeneity: Tau² = 0.06; Chi² = 5.44, df = 4 (P = 0.25); I² = 26%
Test for overall effect: Z = 1.02 (P = 0.31)

| | | | | | | | |
|---|---|---|---|---|---|---|---|
| **2.1.5 NICU Admission** | | | | | | | |
| Bekker 2023 | 18 | 100 | 20 | 100 | 45.9% | 0.90 [0.51, 1.60] | 2023 |
| Monincx 1997 | 18 | 77 | 23 | 74 | 54.1% | 0.75 [0.44, 1.28] | 1997 |
| Subtotal (95% CI) | | 177 | | 174 | 100.0% | 0.82 [0.55, 1.20] | |
| Total events | 36 | | 43 | | | | |

Heterogeneity: Tau² = 0.00; Chi² = 0.20, df = 1 (P = 0.65); I² = 0%
Test for overall effect: Z = 1.02 (P = 0.31)

**Fig 5. Summary risk ratios presented as forest plots for assessed perinatal outcomes.**

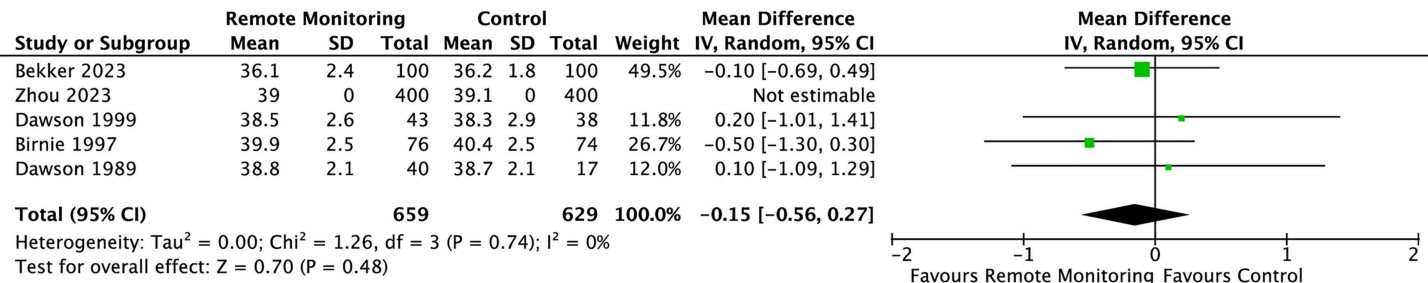

**Fig 6. Summary mean difference presented as forest plots for the assessment of gestational age delivery.**

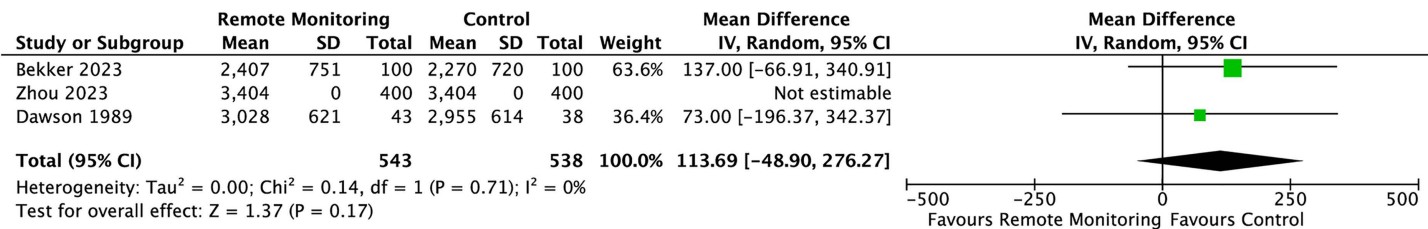

**Fig 7. Summary mean difference presented as forest plots for the assessment of birthweight.**

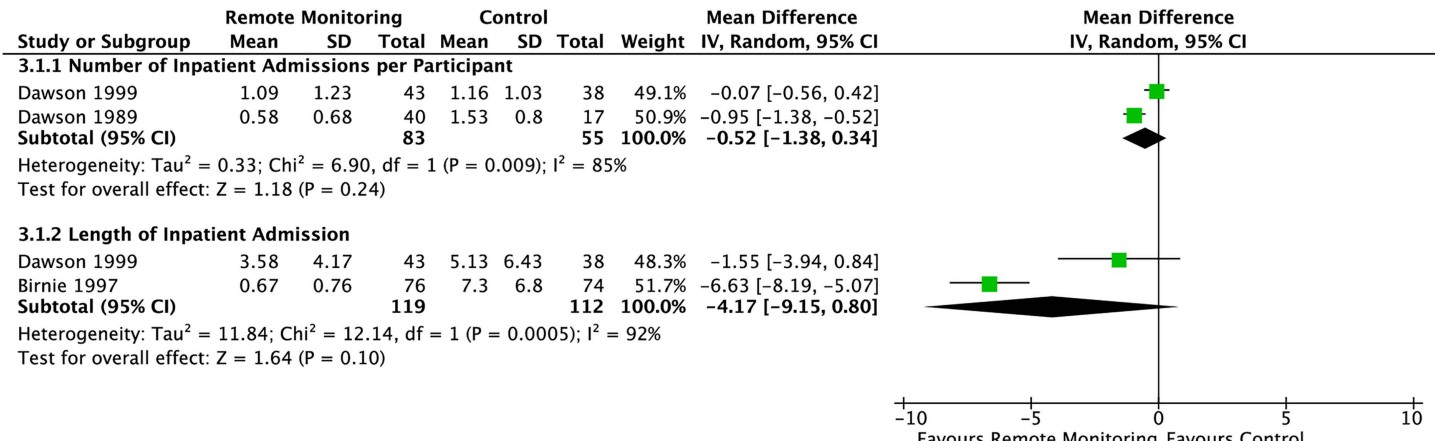

**Fig 8. Summary mean difference presented as forest plots for the assessment of healthcare usage outcomes.**

Albeit, with few studies included in this analysis, the validity of the presented heterogeneity is limited. Regarding, hospital admission and inpatient stay, it is important to note that the control groups in both Dawson et al. and Birnie et al. were randomised to directly to immediate hospital admission, likely explaining the large disparity in results between intervention and control groups in the respective outcomes [30].

Additionally, as expected with remote monitoring, Dawson et al. noted a significant increase in community midwifery visits (3.7 vs 1.4 visits; $p = 0.002$) and longer visits (33.5 vs 12.8 minutes per visit; $p < 0.001$). However, the number of antenatal outpatient visits was not significantly different (2.4 vs 3.2 visits; $p = 0.11$). Finally, Monincx et al. demonstrated a non-significant difference in the duration of postnatal admission between home monitoring and control groups (3.2 vs 3.5

days; p = 0.53). Importantly, no included randomised controlled trials clearly assessed the impact of home CTG monitoring on emergency attendance to maternity triage; an important outcome when evaluating the burden of a monitoring strategy on obstetric services.

## Subgroup analysis

Due to the small number of studies available for each meta-analysed outcome, subgroup analysis could not be performed with respect to monitoring frequency, type of person undertaking the remote monitoring, risk of bias judgement and age of study.

## Observational studies

**Feasibility.** Twenty-three studies primarily evaluated feasibility outcomes in relation to remote CTG usage. Transmission delays and failures were scarcely reported by included studies. In synchronous systems, transmission of remote data was in real-time, however, reporting of delays, albeit likely only to be a few seconds, were not provided by included studies. Synchronous transmission failures of 7% was noted by Gonen et al. whereby 34 participants completed home CTG monitoring up to three times per week which was synchronously tele-transmitted via telephone lines [39]. Amongst asynchronous systems, whereby completed CTG traces were then subsequently transmitted for review, average transmission delays ranged between 30 seconds and 33 minutes [28,36,40]. However, Dawson et al. reported delays of up to two hours in certain cases, typically occurring on weekends [36]. Transmission failures ranged from 1-16%, however, minimal studies commented on this [36,44,48,58,61]. Primarily failure of tele-transmitted CTGs were attributed to insufficiencies in technological infrastructure support (e.g., poor modem operation), remote equipment malfunction and patient handling errors.

Remote CTG quality and interpretation was assessed by several included studies, frequently through subjective assessment by individual reviewers [28,36,37,39–42,44,47,56–58,62,63]. Overall, the ability to interpret remote data was moderate to excellent, ranging from 74-100% of all monitoring episodes, with most studies citing >90% of remote CTGs were suitable for interpretation [28,36,37,40–42,44,56,57,62,63]. Decreasing gestational age at time of monitoring and increased maternal weight were both negatively correlated with remote CTG quality [54,58,61]. Signal loss was reported in a select number of studies, ranging from 4-40% [33,37,39,63]. Unsupervised versus supervised participants were noted to have greater signal loss during home monitoring [37], whilst Gonen et al. demonstrated that repeated monitoring episodes had minimal impact on reducing the rate of signal loss (21% versus 20%) [39]. Methodological criteria for what defined a remote CTG episode as interpretable varied substantially between studies and was frequently undefined by authors. This must be taken into account when considering the above results, as the subjective nature of interpretation may introduce bias into the reported results. However, when authors defined what classified a remote CTG episode as interpretable, this widely varied, from just 40% or 70% of the trace being successfully tele-transmitted for review by Gonen et al. and Uzan et al. respectively [39,58], or requiring 10 minutes of continuous CTG monitoring by Tamaru et al [57]. It is therefore imperative for future studies to have uniform methodological criteria for remote CTG interpretation, as this enables researchers to robustly determine the suitability of a particular home device for antenatal care usage.

The feasibility of remote fetal heartrate monitoring was further assessed using either the validated system usability scale (SUS) in three studies [42,52,53] or the telehealth usability questionnaire (TUQ) in one study [62]. Total SUS and TUQ scores ranged from 76.5-88.1 out of 100 and 6.6 out of 7, indicating good to excellent participant usability. When measured against in-clinic use of the device, scores were comparable to home use [52].

The clinical utility of home CTG monitoring was further investigated within a select number of feasibility studies. Home fetal monitoring enabled early recognition and prompt intervention of fetal concerns within two studies [36,46]. Additionally, Hamm et al. evaluated a synchronous remote monitoring device, incorporating internet tele-transmission in 34 high-risk participants. Interpretability was high (93.9%), which subsequently enabled a reduction in outpatient office visits by 88.5%

[42]. Similarly, Moore et al. demonstrated a reduction in hospital admissions by 53% following the introduction of a remote monitoring system for 100 participants [48].Finally, Axelrod et al. noted a statistically significant reduction in total clinic time for participants undergoing remote monitoring versus in-clinic monitoring (median 59.0 min vs. 159.0 min, P < 0.001) [62]. This was assessed in 20 women complicated with gestational diabetes whom used remote CTG in combination with urine analysis, glucose testing and ultrasound examination at home, followed by a telemedicine consultation [62].

**Diagnostic accuracy.** Just two studies from the same authors primarily evaluated outcomes in relation to the diagnostic accuracy of remote CTG monitoring [52,53]. The reference standard was conventional outpatient CTG analysis, with different recruitment periods noted within each study. Both studies utilised a handheld, lightweight device, employing ultra-wide beam Doppler technology, enabling measurement of the fetal heartrate. The device further integrates a dedicated optical sensor to directly monitor the maternal heartrate from the abdomen, eliminating fetal-maternal heartrate cross-talk. The system includes a smartphone-based interface that displays the fetal heartrate trace and is calculated on a Bluetooth-connected smartphone, which is then uploaded to a clinical management system at the respective clinic. Median/average length of remote monitoring ranged from 4.6 to 28.5 minutes in both studies [52,53].

When comparing individual data points of remote CTG monitoring versus outpatient monitoring in both studies, the mean beat per minute (bpm) difference ranged from -0.3 to 0.63 (95% CI ranging from -4.89 to 5.32), demonstrating superior coincidence [52,53]. Interestingly, participants using an additional belt to help secure the device during monitoring, achieved slightly lower fetal heart rate agreement. However, mean signal loss during remote monitoring was minimal at $6.7 \pm 5.9\%$ [53]. Intraclass correlation was excellent ranging from 0.966-0.99 in both studies, demonstrating remote CTG as a promising tool for supplementing current outpatient CTG monitoring from a diagnostic accuracy perspective. Linear regression analysis of the 2022 results from Porter et al. demonstrated no evidence of proportional bias [53].

**Pilot study.** The usage of remote CTG within the context of a non-randomised pilot study, whereby the control group was standard care, was demonstrated in two studies. These studies used patient-operated mobile CTG devices with tele-transmitted data via internet connectivity [38,51]. Regarding maternal outcomes, there were no significant differences in the rates of caesarean section, preterm birth or low neonatal birthweight. Regarding neonatal outcomes, Pan et al. demonstrated a significantly reduced rate of neonatal asphyxia in those under remote monitoring versus conventional care (1.58% versus 3.18%, p = 0.021) [51], however, this was reported as not significant by Gan et al (4% versus 7%, p = 0.483) [38]. Conversely, Gan et al. noted an increased rate of neonatal intensive care admission in the remote monitoring group (12% versus 8.3%, p = 0.044), however, their composite adverse neonatal outcome, comprising of several adverse perinatal sequalae (neonatal death, fetal distress, admission to NICU) was non-significant (p = 0.319) [38]. Interestingly, several demographic characteristics were evaluated by Gan et al., noting participants recruited to undertake remote CTG monitoring were more likely to be nulliparous (58.0% versus 44.6%, p < 0.001), classify as a high-risk pregnancy (68.8% versus 60.0%, p = 0.003) and to have a higher education level (57% versus 52% undergraduate education, p < 0.001). The reasoning for these disparities is unclear, albeit the authors suggest a greater proportion of nulliparous participants undertook remote monitoring as they may have a reduced sense of pregnancy fear compared with multiparous participants [38]. Furthermore, education level is closely linked to digital literacy and is a perceived barrier to telehealth update, which may explain why women with lower education levels were less prevalent in such studies [64].

Responses from participants noted that those assigned to the remote monitoring group demonstrated a more comprehensive knowledge of fetal heart rate monitoring, including appropriate interval timing, whilst further highlighting a significant reduction in anxiety scores using the self-rating anxiety score (SAS) [38,51]. Despite over 3900 participants recruited between both studies, the evaluation of system-based outcomes was minimal to obsolete. Soley, Pan et al. noted a slight reduction in hospital attendances in the remote monitoring group, however, this was not significant (7.86 versus 8.13, p = 0.283) [51]. Further prospective studies are encouraged to evaluate system-based outcomes alongside clinical outcomes, for a more comprehensive analysis of remote telemonitoring interventions.

**Device safety.** When considering use of a technology with the potential to act as an adjunct or supplement to traditional antenatal care, it is vital to consider the safety provisions employed to manage device related concerns. Unfortunately, only 28% of included studies provided such information [9,36,39,41,45,54-56,58,60,62]. Brief statements were provided in five studies, whereby if remote CTG's demonstrated abnormal findings, then attendance to hospital was advised [9,36,54,58,62]. Six studies supplied additional information referring to specific CTG recording abnormalities which would classify a remote monitoring episode as abnormal. However, this remained variable in reporting studies, ranging from just the absence of accelerations [39,45], to recurrent variable decelerations and tachycardia [56], or a comprehensive list of possible abnormal features relating to each assessed aspect of the CTG recording (baseline heartrate, variability, decelerations, accelerations) [41,55,60]. However, just two studies provided sufficient information regarding what the hospital assessment entailed, detailing either a repeat outpatient CTG and/or a sonographic assessment of the fetus [55,60]. Consequently, it is vital for future studies to improve on the adequacy of methodological detail relating to remote CTG safety concerns, ensuring researchers are able to sufficiently replicate available studies.

The median number of remote CTG's which demonstrated abnormal findings was 6.5% (range 0–36%), albeit, 80% of studies had rates of 8% or lower [36–38,42,45,48,50,54–56]. It should be noted that the single study with 36% abnormal remote traces, only included participants with isolated structural abnormalities, thus the rate of abnormal CTGs is likely to be far greater. Frequently, abnormal monitoring episodes were repeated at home or in hospital and were subsequently normal. Adverse outcomes relating directly to the usage and the technical aspect of the equipment were rarely explored in the included studies. However, when noted, no adverse outcomes occurred in relation to research involvement [62].

**Patient and provider experience.** Four of the 37 aforementioned studies primarily evaluated patient and/or provider experiences with remote CTG monitoring, either through focus group or a semi-structured interview format [11,27,29,43]. Patients frequently favoured home monitoring over hospital monitoring, often due to a reduction in travel time, increased convenience and the positive effect this had on family life [11,43]. Patients acknowledged how home monitoring can extend the patient-clinician relationship, whilst enabling patients to take an active role as their own care provider during pregnancy [27,29]. Whilst empowering, participants also recognised the challenges associated with remote monitoring, frequently attributed to the rise in clinical responsibility for one's health [29]. Additional studies examined patient satisfaction using either a Likert scale [34,41,45,57], or yes/no answering [54,63], demonstrating high user satisfaction. Furthermore, participants cited that remote monitoring devices were easy to use and provided reassurance [44,48,62,63].

Home monitoring was perceived by stakeholders as a more comfortable and less stressful environment for patients, whereby these positive attributes would be most beneficial for certain high-risk groups requiring regular monitoring (hypertensive disease in pregnancy) [27]. Clinicians further cited benefits of home monitoring in relation to improving patient control and understanding of their own pregnancy. However, certain obstetricians and midwives did express fear of losing overall control over the patient's pregnancy management, recommending that a wider change in clinical perspective may be required to enable the synergy between home and hospital care [27]. Furthermore, stakeholders highlighted patient safety concerns regarding remote monitoring, but emphasised that in-depth patient training, regular patient-provider communication and synchronous CTG reviews could help manage these concerns. Irrespective, it was acknowledged that patients would likely feel a greater sense of safety at home, which is an important factor to consider within the global assessment of home monitoring [27].

**Economic burden.** Required capital for implementation and continued costings for maintaining remote CTG services are vital considerations. No included studies were primarily economic evaluation studies, however, several observational studies and randomised controlled trials described or analysed cost outcomes alongside clinical outcomes [9,30,32–34,39,41,48,49]. Initial startup costs were variable, but ranged from £1524 for a single remote device to £52,000 for a multisite set up, which when adjusted for inflation to today's current cost, would be approximately £98000 [34,39,41,48]. However, universally, included studies demonstrated consistent antenatal non-fixed savings, ranging from £1552 to £2113 per participant within the remote monitoring groups, enabling swift recovery of initial implementation

costs [9,30]. Frequently savings were as a result of fewer antenatal admissions [9,30,32–34], in addition to a consistent reduction in the cost and number of telemedicine encounters versus regular outpatient appointments [34,49]. Whilst Dawson et al. did note that community midwifery time and respective travel costs may increase for participants undertaking remote monitoring, there was also a reduction in time off work for both participants and partners who were randomised to domiciliary care [34]. When costs were collated with antenatal clinic visits, inpatient admission days, alongside the initial infrastructure costs, remote CTG monitoring was £223.83 cheaper than conventional care per participant for antenatal management. Therefore, economically, remote CTG monitoring may be a viable option for antenatal care, however, it is important that the initial startup costs are mediated by ensuring the service is offered to a wide enough number of women and for an extended duration to maintain viability.

**Risk of bias.** The detailed results of the risk of bias assessments for all included articles are presented in S6 Table. A summary of the risk of bias results for the seven randomised controlled trials using the Cochrane RoB 2 tool can be seen in Fig 9. The overall quality of studies was either moderate (three studies) [9,34,47] or high risk of bias (four studies) [30,35,59,60]. This was primarily attributed to both participants and researchers' awareness of the assigned intervention during the trial, which is an inherent bias due to the technology being assessed. However, the majority of studies did not provide adequate information to determine if an appropriate analysis was used to estimate the effect of assignment to intervention, thus determining a higher risk of bias. Furthermore, studies frequently did not include a pre-specified analysis plan prior to the production of results, leading to some concerns of bias surrounding the selection of reported results. Summary results for the 28 observational studies using the modified QUADAS-2 checklist are presented in Fig 10. The overall quality of the articles was heterogeneous, with moderate to high risk of bias. This was primarily attributed to the suboptimal quality of methodological reporting with respect to participant inclusion/exclusion criteria, inadequate

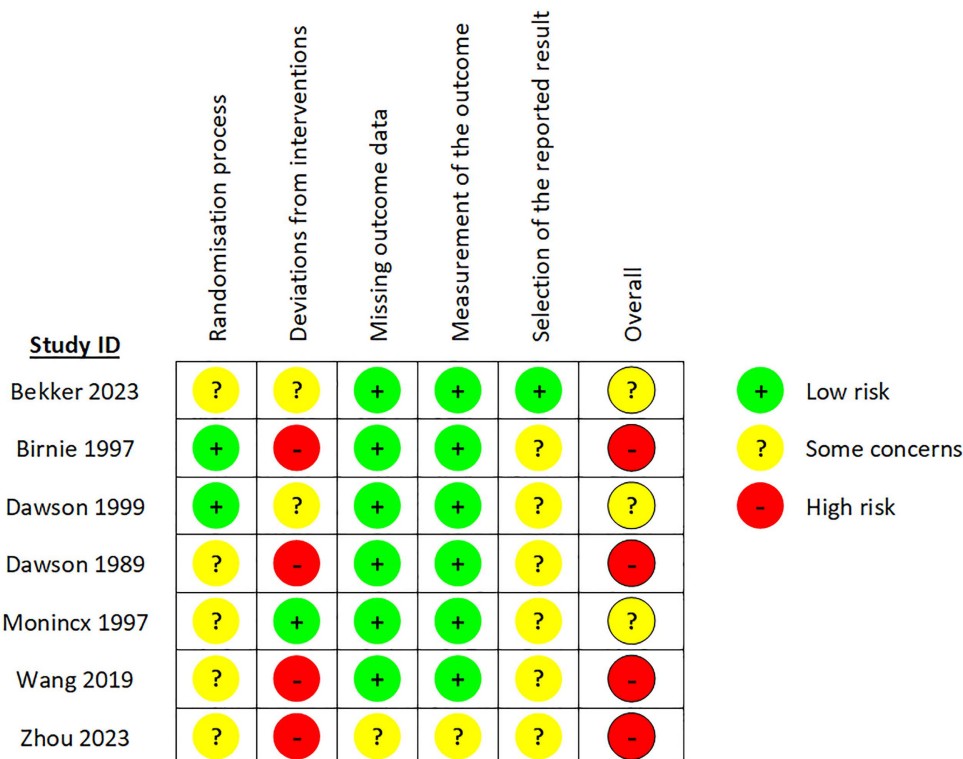

**Fig 9. Risk of bias results for randomised controlled trials using the Cochrane risk of bias (RoB 2) tool.**

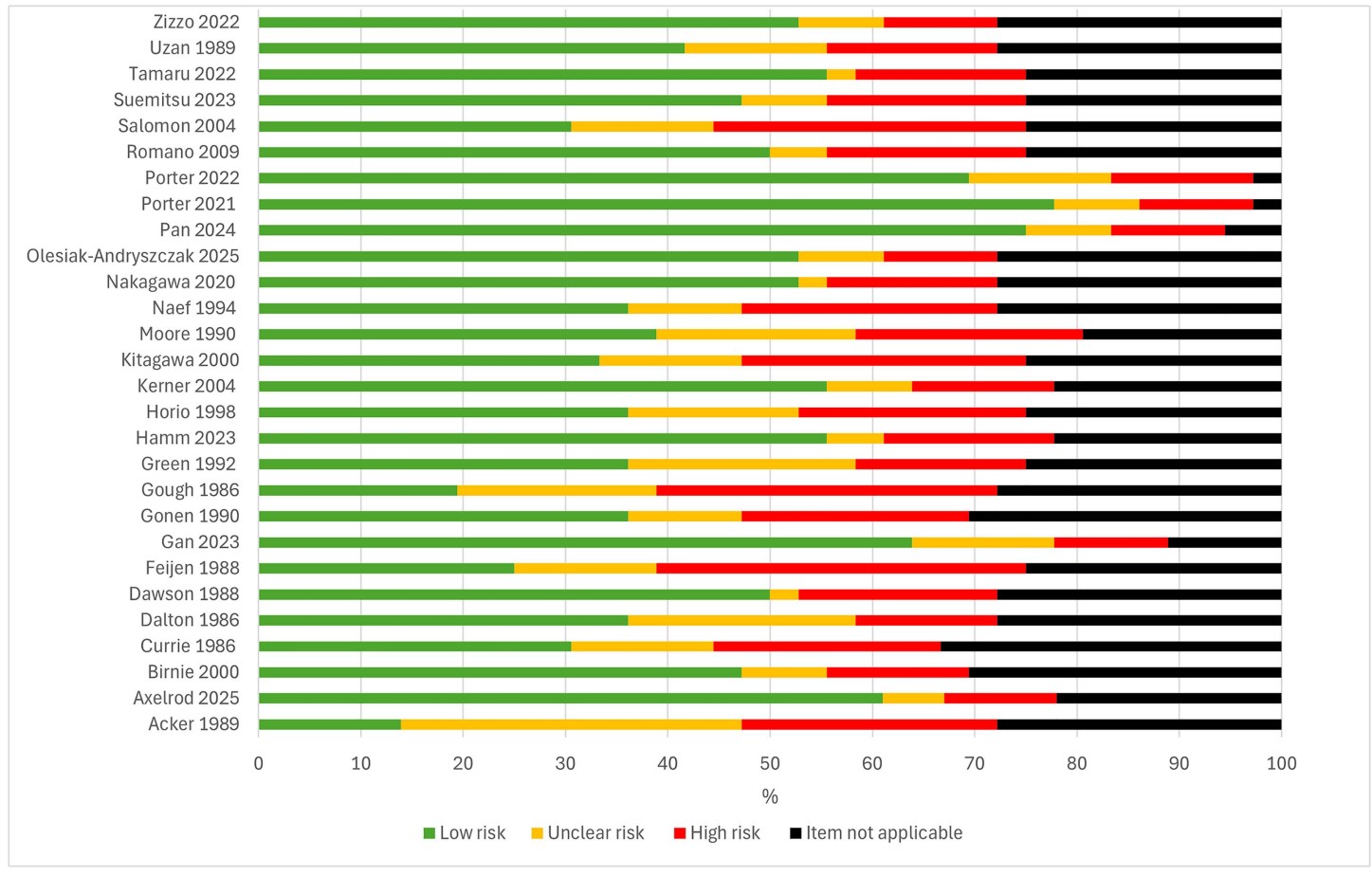

**Fig 10. Risk of bias results for clinical observational studies using the QUADAS-2 checklist.**

reasoning for sample size and insufficient detail to enable study replication. Many studies lacked a reference group or comparison, restricting researchers to determine quantitative comparisons to standard care. Most commonly, included observational studies failed to provide information regarding cost implications. When provided, only brief cost statements were usually provided [32,33,39,41,48,49,53], with the exception of one study providing slightly more detailed crude cost estimations [61]. Cost assessment was more robust within included randomised controlled trials. However, it should be noted that overall risk of bias was generally lower in more recent versus older published studies. Sole qualitative studies were generally of lower bias risk across all categories assessed within the JBI frameworks, however, frequently there was unclear or inadequate information provided relating to the theoretical and cultural standpoint of the researcher.

A summary of study findings in this review, including recommendations can be viewed in Table 3.

## Discussion

This systematic review and meta-analysis was designed to comprehensively assess the role of home CTG monitoring for obstetric care, by evaluating its impact on clinical outcomes, including assessing the feasibility, diagnostic accuracy, acceptability, utility and economic burden. Despite included studies demonstrating a high risk of bias, primarily due to methodological heterogeneity, remote CTG monitoring demonstrated non-inferiority across all maternal and perinatal outcomes. Synchronous and asynchronous remote CTG monitoring systems were feasible in a wide range of settings and

**Table 3. Summary of existing evidence.**

| Study type | | | | |
|---|---|---|---|---|
| **Randomised controlled trial** | **Feasibility** | **Diagnostic accuracy** | **Pilot** | **Qualitative** |
| **Evidence**: There was no difference in maternal or perinatal outcomes between those randomised to remote CTG monitoring, versus conventional care. Random effects analysis demonstrated no change in inpatient admission and length of inpatient stay between groups. Included studies demonstrated moderate to high risk of bias.<br><br>**Recommendation:** Further high-quality research is required with additional in-depth assessment of service-led and cost factors, in conjunction with clinical outcomes. | **Evidence**: Feasible to transmit remote CTG data synchronously and asynchronously with moderate to excellent interpretability. However, frequently assessment of data was based on subjective factors alone and there was a lack of uniform methodological criteria for interpretation. Transmission failures and delays were more apparent in asynchronous systems. Usability was rated highly among participants.<br><br>**Recommendation:** Greater methodological detail is required in future studies, with increased documentation on transmission details for synchronous systems. | **Evidence:** Remote CTG monitoring was comparative to conventional outpatient CTG, demonstrating superior coincidence. Intraclass correlation was excellent.<br><br>**Recommendation:** Minimal recommendations required. However, there were only two studies included in this review, therefore further studies, including greater numbers of women are encouraged. | **Evidence:** Most clinical outcomes were not significantly different between remote monitoring and control groups. Initial insight into a potential trend to telehealth uptake in participants of some demographic groups.<br><br>**Recommendation:** Further prospective pilot studies, particularly including the assessment of service-led outcomes are warranted. | **Evidence:** Remote CTG monitoring was associated with improved patient satisfaction, due to reduced travel requirements. Service users cited benefits regarding improved patient control and understanding. Concerns were raised over safety issues, which may be mediated with in-depth device training.<br><br>**Recommendations:** Minimal recommendations required, albeit further insight into provider viewpoints would prove beneficial. |
| **Additional outcomes** | | | | |
| Device safety | | Economic evaluation | | |
| **Evidence**: Minimal methodological reporting of safety protocols in included studies. When included, information provided was variable between studies, with very few including information of what the hospital assessment included. Rate of abnormal remote monitoring traces was generally low, and device related adverse outcomes was scarcely reported.<br><br>**Recommendation:** Future studies warrant greater methodological information on safety protocols, including hospital assessment details and subsequent safety outcomes. | | **Evidence:** Initial startup costs were high but frequently reclaimed back over time following implementation due to non-fixed cost savings such as fewer antenatal admissions. Remote monitoring appointments frequently cost less than standard care appointments.<br><br>**Recommendation:** Full health economic analysis required, likely in conjunction with a randomised controlled trial. | | |

the ability to interpret remote data was graded as moderate to excellent in reporting studies. However, frequently reviewer interpretation was subjective, with a variety of different definitions for when a CTG was classified as interpretable, which highlights a potential concern for bias. Transmission failures were generally low, but when occurring were frequently in relation to infrastructure and equipment errors. Transmission delays were infrequently reported, albeit more commonly cited in asynchronous systems, with periods of extended delay. Transmission delay is an important aspect to consider when minimising safety concerns relating to remote monitoring, particularly in high-risk patient groups whereby emergency intervention following fetal monitoring may infrequently occur. Therefore, remote CTG devices capable of synchronous (live interpretation) rather than asynchronous tele-transmission may be more suitable to assess in future research. The diagnostic accuracy of remote CTG monitoring was promising, with comparative capabilities to conventional outpatient CTG with respect to coincidence and beat-to-beat variability.

Overall acceptability ratings for remote CTG monitoring were high for both patient and providers, citing common benefits in relation to satisfaction, reduced travel, economic savings, increased access to obstetric care and balance in healthcare equity. Whilst implementation and infrastructure costs for remote CTG monitoring were high, costs could normally be accrued back over time due to non-fixed savings, such as consultation and travel costs from the remote service versus routine care. Methodological reporting of device safety was infrequently reported, with minimal information on specific CTG criteria warranting attendance to hospital or subsequent detail about hospital assessment. However, generally the rate of abnormal remote CTGs was less than 10% and adverse events relating to the technological capabilities of the

devices was minimal. Additionally, information regarding service-led outcomes such as hospital attendance/admission and the impact on clinical workflow were scantly reported, however, are vital outcomes to assess. These outcomes are essential for evaluating the real-world burden and benefit of remote CTG. There was an underrepresentation of pilot and high quality randomised controlled trials, suggesting a need for further robust research. Furthermore, current included literature provided limited reporting of clear methodological, safety pathways and technological capabilities of the remote CTG systems, proving difficulty for researchers to accurately replicate studies. Current evidence is graded as low or very low using the GRADE approach, suggesting future studies should consider a more rigorous approach to study design, in order to develop a greater cohesive and applicable body of evidence to advocate for routine implementation in standard antenatal care pathways.

The meta-analysis findings of this review align similarly to the only prior published review by Li et al., which explored the effectiveness of remote CTG monitoring in nine randomised controlled trials [13]. Maternal and perinatal outcomes were not significantly different between remote and usual care groups. However, Li et al. additionally demonstrated a statistically significantly reduced rate (34%) of neonatal asphyxia, defined by low Apgar scores (<7), in the remote care group. Upon closer inspection of this review, it can be noted that included studies evaluated remote CTG monitoring in a range of settings and pregnancy stages. This included combining remote antenatal assessment at home alongside intrapartum examination in hospital [13]. Amalgamation of studies for meta-analysis from different pregnancy stages warrants some concern, as it poses difficultly to clearly establish which area remote CTG monitoring is most beneficial. Particularly given the wide variation of potential management strategies in the antenatal and intrapartum stages. In our updated review, which specifically focusses on home antenatal monitoring, we demonstrated a non-significant difference for neonatal asphyxia. Similarly to our review, a lack of high-level evidence was available for inclusion highlighting a clear need for an in-depth evaluation of remote monitoring on clinical maternal-fetal outcomes.

As previously noted, many included studies contained a small number of participants, which can be typical when assessing digital monitoring interventions [65]. However, in the context of a randomised controlled trial this can be troublesome for sufficiently evaluating clinical outcomes resulting in severe consequences, such as perinatal mortality. Therefore, employing a compositive adverse outcome score may prove beneficial in future research to evaluate the effectiveness of home CTG. Only, one included trial had utilised this, incorporating several outcomes such as perinatal mortality, low Apgar score, admission to NICU and caesarean section [9]. Presently, there is a lack of clear consensus regarding which clinical and service-led outcomes should be integrated into a composite outcome. The development of a core outcome set for remote obstetric studies may prove beneficial to reduce any inconsistent outcome reporting and improve the evidence for remote CTG research.

Aside from remote CTG monitoring, prior reviews have demonstrated the effectiveness and feasibility of several other obstetric telehealth interventions and remote/wearable monitoring devices, taking into consideration a range of clinical and service-led outcomes [66–69]. Consequently, in an era of heightened technology development, the diverse availability of digital healthcare interventions is promising and promotes an innovative landscape in obstetric healthcare. This is pertinent area to explore given the ever-growing capacity concerns within obstetric services, secondary to an ageing maternal population and increased classification of high-risk pregnancies which warrant serial antenatal monitoring, including CTGs [1,70]. Remote CTG monitoring, in conjunction with additional monitoring strategies, such as blood pressure, maternal heartrate, glucose monitoring and maternal temperature may be a potential avenue to modify current antenatal care models and improve the streamlining of outpatient services. In cases where high-risk patients are required to attend the hospital multiple times a month for CTG monitoring, such as in preterm prelabour rupture of membranes, the incorporation of a home-hospital hybrid antenatal CTG model may provide relief on maternity outpatient services. However, further evaluation is still required, particularly regarding the impact remote monitoring has on healthcare usage, such as unexpected inpatient admission and emergency outpatient attendance, as well as cost effectiveness. Given the research difficulty in assessing the beneficial impact of home CTG on reducing perinatal mortality, as is already seen in the case

of conventional CTG, home CTG is likely to need to demonstrate superiority against these healthcare usage and economic outcomes, including acceptability compared with conventional CTG to even consider implementation. Technological aspects such as data security are equally as important to evaluate, which were significantly under-evaluated in included studies and have also been highlighted as a research gap in prior tele-health reviews [69]. Furthermore, a clear understanding of the training requirements needed for women to adequately perform the monitoring requires further evaluation and is not currently clear in the available literature. Finally, the implementation of a new antenatal technology should aim not to overburden, but rather support healthcare professionals, in what is an already busy clinical environment. Clear support networks for technology issues, safety pathways, improvements patient flow and reduced digital complexity can all aid in improved clinician satisfaction and acceptance. Finally, choosing which high-risk patient groups to undertake remote CTG monitoring versus conventional care is still unclear. The emergence of predictive models for stratifying risk of poor outcomes, such as placental growth factor testing for pre-eclampsia, may be of further use in informing which patient groups may benefit from only in-person care versus groups eligible for home care [71]. Predictive tools may assist in the management of safety concerns with remote CTG monitoring, by stratifying the risk of adverse maternal-fetal outcome. Enhancement to antenatal healthcare models is currently a recommendation for research by the National Institute for Health and Care Excellence (NICE), suggesting a national focus on developing UK healthcare services digitally, whereby remote CTG may be an avenue [72].

An important aspect to consider is the close relationship between technological advancement and the economic state of the respective country [68]. Whilst remote monitoring may improve patient access to care, CTG equipment in particular can be expensive for the healthcare provider, whilst potentially requiring the patient to have established home infrastructure (e.g., Wi-Fi connectivity) in order to use the device. This is evidenced as included studies in this review solely derive from high-income counties. Crucially, these factors, amongst many others, such as availability of electricity and costs of device maintance, can be barriers for implementation in low- and middle-income countries [73]. However, given the significantly higher rates of perinatal mortality in these countries, the need for remote fetal monitoring may be more urgent and thus future research in this area is strongly encouraged [13,74]. The develop of a low-cost remote device, capable of tele-transmission over low bandwidths may aid to reduce the geographical and global disparities seen in obstetric antenatal care.

Crucially, the endorsement of remote monitoring devices, such as CTG, stem from clinician attitudes regarding implementation. Initial insights from this review highlight generally high provider acceptability, suggesting a potential turning point in remote fetal monitoring adoption [10]. However, concerns relating to the safety of remote monitoring were noted, ensuring participants were fully educated on device usage. An important consideration regarding remote monitoring implementation is participant digital literacy. Whilst remote CTG monitoring may improve access to care, it is well-known that digital literacy is lower in those from most deprived groups [64]. Unfortunately, participants with low educational levels only made up between 4–8% of the remote monitoring cohorts in the included pilot and randomised controlled trials. Consequently, it is imperative that future research supports inclusivity of all patient groups, ensuring robust training strategies to enable a comprehensive understanding of device usage and thus maximise digital literacy.

Finally, the environmental impact of home digital technologies, such as CTG need to be considered when implementation occurs on a wider scale. Direct benefits of home CTG may include a reduction in travel and potential reduced hospital attendance, thus limiting patient transportation emissions and resource use in hospital respectively. However, one must also consider the environmental impact of device manufacturing such as the use of metals (e.g., lithium), plastics (e.g., petroleum-derived) and the associated energy-intensive manufacturing processes. Furthermore, data transmission requires server power and network infrastructure, which on a large scale may contribute to healthcare's growing digital carbon footprint. Ideally, a reduction of single use items will reduce the environmental impact associated with remote CTG. However, items such as gel and abdominal belts are unlikely to change from single use, similar to current in-hospital monitoring, In an era of environmental scrutiny, home monitoring devices, such as CTG, need to demonstrate environmental benefits to assist with wider implementation.

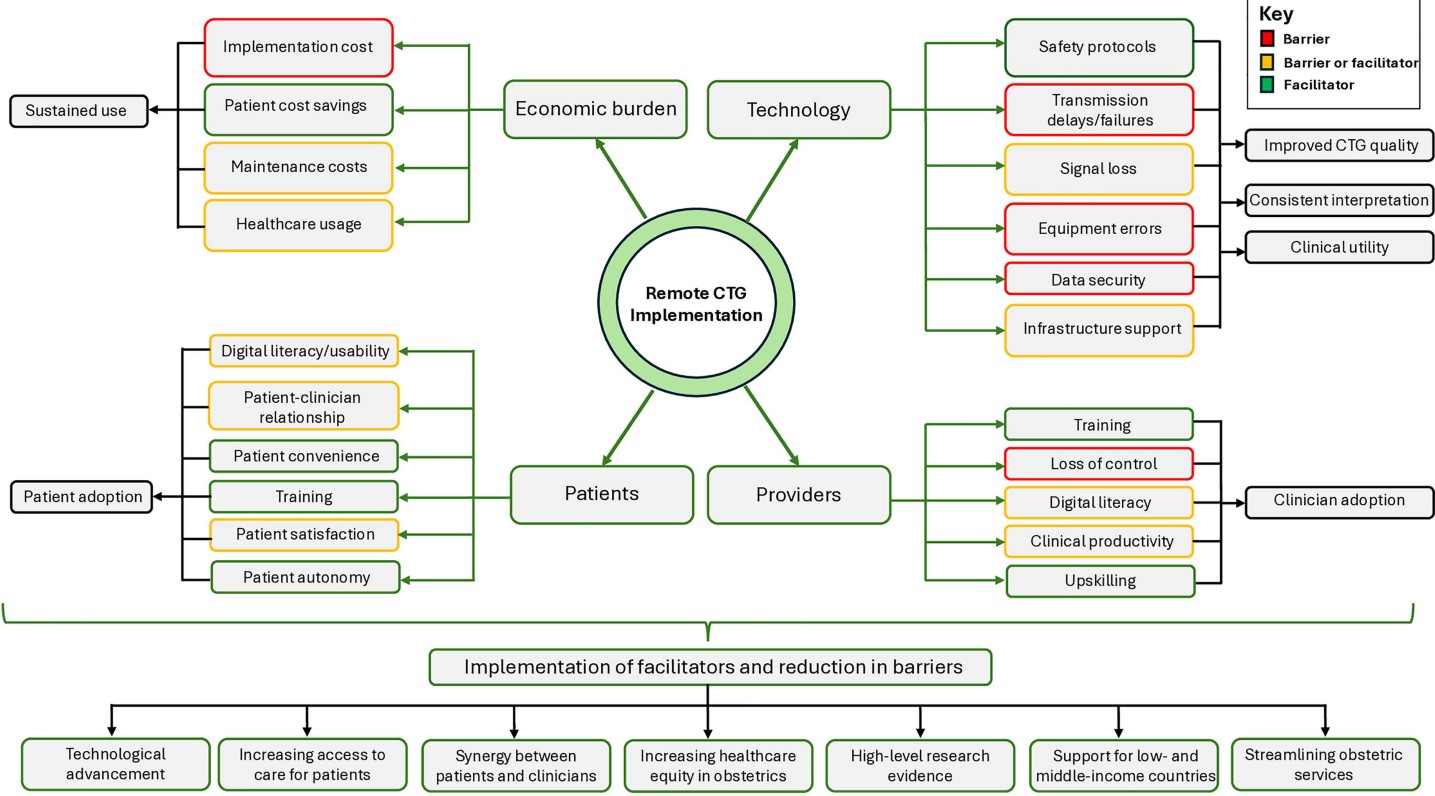

Fig 11 represents a selection of barriers and/or facilitators for remote CTG implementation, which have been stratified from the evidence consolidated in this review. Minimisation of barriers and adoption of facilitators may enable clinicians to achieve important beneficial outcomes listed at the bottom of the figure.

The extensive search, including grey literature and subsequent evaluation of a wide range of outcomes, including meta-analysis, represent the main strengths of this review. To our knowledge, this review is currently the most comprehensive assessment of remote CTG monitoring for obstetric care. Importantly, this review enables clinicians and researchers to understand the current state of home CTG monitoring and determine how best this technology can be safely implemented into future antenatal care. This review further facilitates clinicians to consider the technological requirements for optimising remote monitoring, whilst determining the correct elements required for robust future research.

With regards to limitations, the number of randomised controlled trials was minimal and limits the data which was eligible for meta-analysis and subsequent subgroup analysis. Additionally, included studies were frequently at high-risk of bias, with insufficient methodological reporting to enable complete study replication. These factors should be considered when interpreting the presented results. Furthermore, due to the methodological heterogeneity of included observational studies, a narrative review was required to consider feasibility, acceptability, economic and service-led factors. Given that these outcomes are integral to appropriate technology implementation, it is suggested that future studies clearly delineate such factors to enable a thorough quantitative analysis of remote CTG monitoring. Additionally, many included studies were relatively old, which may reduce the generalisability of our findings. This relates particularly to the frequent usage of telephone transmission for remote CTG, which is now obsolete in most high-income countries, following the development of internet digital technology transmission. In addition, all included studies were from high-income countries, whereby

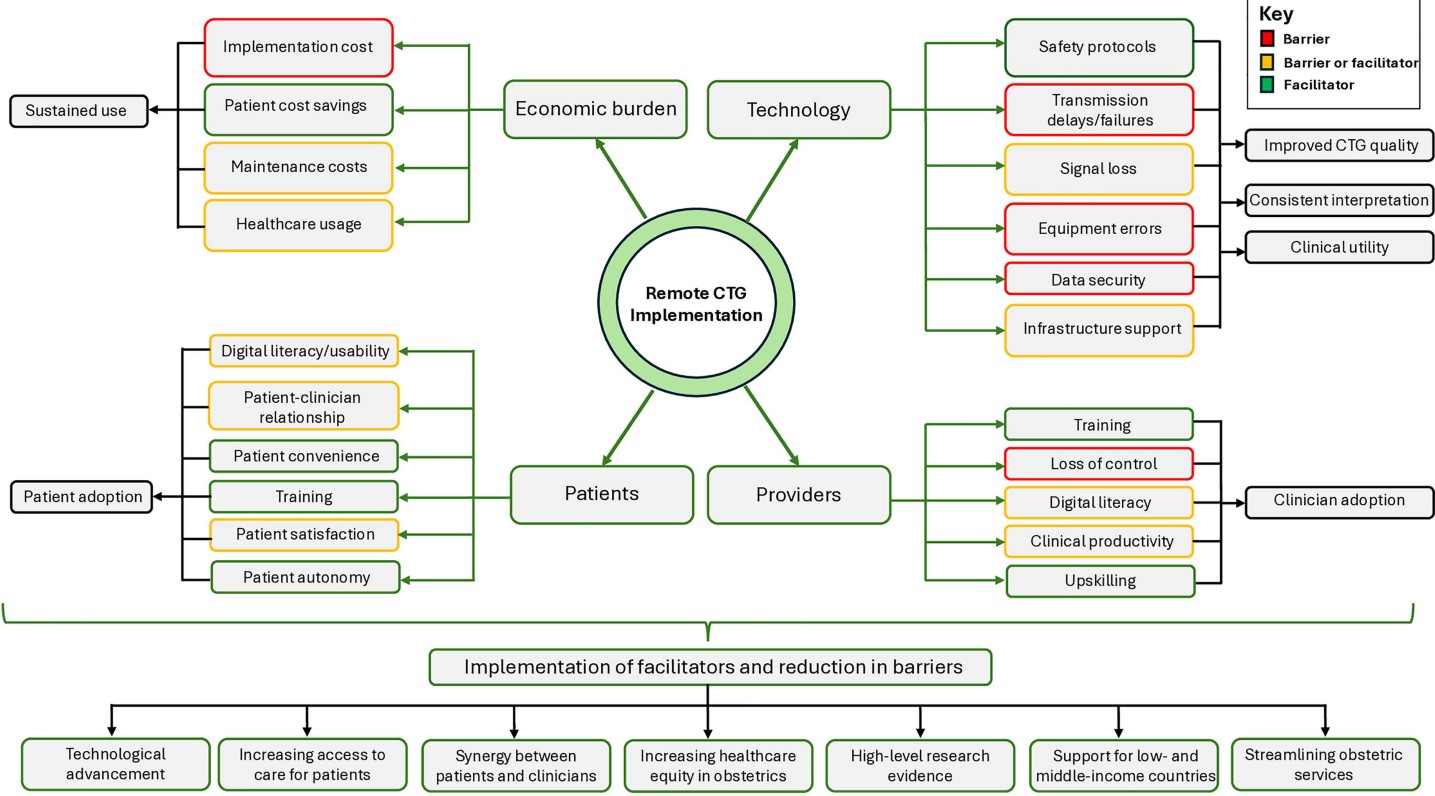

**Fig 11. A schematic diagram illustrating several barriers and facilitators to achieving successful remote CTG implementation.**

available capital for technological infrastructure is likely greater, compared to low- and middle-income countries, thus limiting the global applicability of this review.

## Conclusion

This systematic review and meta-analysis aimed to explore the potential effectiveness, applicability and value of remote CTG monitoring for obstetric care across a wide range of outcomes. Home CTG demonstrates noninferiority to conventional antenatal care on a wide range of clinical outcomes and represents a promising avenue for antenatal care management. However, current evidence is of low or very low quality and presently, additional high-quality evidence with sufficient methodological detail and standardised outcome assessment is required prior to making definitive recommendations. However, remote CTG monitoring has the potential to alter current antenatal care models and shape the landscape of digital healthcare innovation in obstetrics.

## Supporting information

**S1 Table. PRISMA checklist.** From: Page MJ, McKenzie JE, Bossuyt PM, Boutron I, Hoffmann TC, Mulrow CD, et al. The PRISMA 2020 statement: an updated guideline for reporting systematic reviews. BMJ 2021;372:n71. https://doi.org/10.1136/bmj.n71. This work is licensed under CC BY 4.0. To view a copy of this license, visit https://creativecommons.org/licenses/by/4.0/.
(DOCX)

**S2 Table. Search Strategy.**
(DOCX)

**S3 Table. Critical appraisal checklist for observational studies.**
(DOCX)

**S4 Table. Excluded studies and associated reasons for exclusion.**
(DOCX)

**S5 Table. Description of study methodology.**
(DOCX)

**S6 Table. Critical appraisal results for all studies.**
(DOCX)

## Author contributions

**Conceptualization:** Jack Le Vance.

**Data curation:** Jack Le Vance, Adekunle Adeoye, Rebecca Man, Nashwa Eltaweel.

**Formal analysis:** Jack Le Vance, Adekunle Adeoye, Rebecca Man, Nashwa Eltaweel.

**Investigation:** Jack Le Vance, Adekunle Adeoye, Rebecca Man, Nashwa Eltaweel.

**Methodology:** Jack Le Vance, Adekunle Adeoye, Rebecca Man, Nashwa Eltaweel.

**Supervision:** Leo Gurney, R.Katie Morris, Victoria Hodgetts Morton.

**Validation:** Leo Gurney, R.Katie Morris, Victoria Hodgetts Morton.

**Writing – original draft:** Jack Le Vance.

**Writing – review & editing:** Jack Le Vance, Adekunle Adeoye, Rebecca Man, Nashwa Eltaweel, Leo Gurney, R.Katie Morris, Victoria Hodgetts Morton.

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
