## [Decision Letter · Decision Letter 0]

8 Nov 2025

Response to Reviewers
Revised Manuscript with Track Changes
Manuscript
**Journal Requirements:**
**Additional Editor Comments (if provided):**
**Reviewers' Comments:**

**Comments to the Author**

1. Does this manuscript meet PLOS Digital Health’s publication criteria?

Reviewer #1: Yes

Reviewer #2: Yes

Reviewer #3: Yes

Reviewer #4: Yes

2. Has the statistical analysis been performed appropriately and rigorously?

Reviewer #1: N/A

Reviewer #2: Yes

Reviewer #3: Yes

Reviewer #4: Yes

3. Have the authors made all data underlying the findings in their manuscript fully available (please refer to the Data Availability Statement at the start of the manuscript PDF file)?

Reviewer #1: Yes

Reviewer #2: Yes

Reviewer #3: Yes

Reviewer #4: Yes

4. Is the manuscript presented in an intelligible fashion and written in standard English?

Reviewer #1: Yes

Reviewer #2: Yes

Reviewer #3: Yes

Reviewer #4: Yes

Reviewer #1: The topic is highly relevant given the increasing demand for fetal monitoring in high-risk pregnancies and the growing integration of digital health technologies in obstetric practice.

Strengths:

1. The review is comprehensive, covering a wide range of outcomes including clinical, feasibility, diagnostic accuracy, qualitative, and economic aspects.

2. The inclusion of both RCTs and observational studies provides a holistic view of the current evidence.

3. The meta-analysis was appropriately conducted, and the narrative synthesis of heterogeneous observational studies is clear and informative.

4. The authors appropriately highlight the need for standardized outcome reporting and improved methodological rigor in future studies.

Major Concerns:

1. Low Quality of Evidence:

The GRADE assessment indicating low or very low certainty for all outcomes is a significant limitation. The high risk of bias in many included studies and methodological heterogeneity undermine the strength of the conclusions.

2. Safety Reporting:

As a clinician, I find the lack of detailed safety protocols and management pathways for abnormal remote CTG traces concerning. Only a minority of studies described follow-up actions, which is critical for clinical implementation.

3. Generalizability:

All included studies are from high-income countries, limiting the applicability of findings to low-resource settings where remote monitoring could have substantial impact.

4. Service-Related Outcomes:

Key outcomes such as emergency department visits, unscheduled hospital admissions, and impact on clinical workflow are underreported. These are essential for evaluating the real-world burden and benefit of remote CTG.

Minor Points:

1. The definition of "interpretable" CTG traces varied widely across studies, which may affect the feasibility conclusions.

2. Subgroup analyses were planned but not performed due to limited studies, which is understandable but limits the exploration of heterogeneity.

3. Recommendations:

4. Emphasize the need for future RCTs with robust design, clear safety protocols, and standardized outcome sets (e.g., a core outcome set for remote fetal monitoring).

5. Encourage studies from low- and middle-income countries to assess feasibility and impact in diverse settings.

6. Include a clearer discussion on how remote CTG could be integrated into existing antenatal care pathways, including training requirements and clinician acceptance.

Conclusion:

This review provides a valuable synthesis of the current evidence on remote CTG and highlights its potential as a non-inferior alternative to conventional care. However, the low quality of evidence and limited safety reporting preclude strong clinical recommendations at this time. With the suggested revisions, this manuscript will be a significant contribution to the field.

Reviewer #2: Abstract – Suggest that the statement that “GRADE assessments were low/very low” is expanded to state “GRADE assessments were low/very low quality of evidence”.

Author summary – I would suggest rewording the first sentence as it currently sounds as though women self-classify as high-risk and I would specify that this relates to antenatal fetal heart rate monitoring as opposed to intrapartum fetal heart rate monitoring – I think that this placed a demand on maternity rather than obstetric services, as at least in the UK context the majority of these monitorings occur in antenatal day units which are midwifery-led. e.g. “In an era where the number of pregnant women classified as high-risk is increasing, the demand for regular antenatal fetal monitoring has subsequently surged which has increased the number of consultations in outpatient maternity services.”

Introduction – suggest altering the first sentence to “Cardiotocography (CTG) is a commonly used investigative modality in maternity care to evaluate fetal condition; it is frequently used in pregnant women classified as at high-risk.” Is it possible to say at high risk of what? Presumably perinatal mortality?

Re reference 1. Does the office of national statistics data cited include information on high or low-risk status? It certainly doesn’t include the possible reasons outlined in the sentence. It would be good to support this statement with evidence, or make it more clear that this is the authors opinion.

In line 74, do the authors think it would be reasonable to add the environmental impact of frequent attendance to hospital? The NHS has a plan to reach net zero by 2040, so this is arguably also a relevant impact of frequent episodes of antenatal fetal monitoring.

I think it would be good to include a justification why the authors have focussed on the non-inferiority of home CTG to conventional care with respect to clinical and service-led outcomes. Could they make their argument clearer? e.g. “For home CTG monitoring to be an acceptable alternative for service users and clinicians it must have equivalent or non-inferior performance in the identification of fetal compromise and superior participant experience and/or cost-performance.”

Methods Line 126 – Should this state the Cochrane Database of Clinical Trials? Medline and PubMed search the same database, but the authors did not search Google Scholar or CINAHL. I think the nursing and midwifery database may have had more information about patient experience. I wonder whether it would be worth including these two other sources of evidence.

Lines 163-165 – I think it would be helpful to understand what the authors assessed in terms of feasibility and clinical utility. Does this mean the proportion of home CTG traces which were recorded or the frequency with which a woman had to attend the maternity unit for a CTG? Or did the authors accept whatever the definition for these outcomes was used in the primary study? I think it would be helpful to add some more detail here.

Line 178 – data is a plural, therefore it should read “Continuous outcome data were compared…”

Line 202 – the authors state the “database and grey literature searches…” but they do not describe any grey literature searches in the methods section.

Line 227 – should this read “11 studies not disclosing the gestation of monitoring.”

Line 300 – I don’t understand what the authors mean by “Results were moderately attributed by a single study”. Do they mean “Results were moderately affected by a single study…” Can the authors clarify this sentence?

In multiple places in the results section the authors state “there is a reduction, albeit not statistically significant…” This could reflect their bias or original expectation regarding the results. I would suggest being more careful with language and simply state that there was no statistically significant difference between the two groups.

I also wonder about the validity of the I-squared statistic with such a small number of studies. https://bmcmedresmethodol.biomedcentral.com/articles/10.1186/s12874-015-0024-z I wonder whether the authors should either state that there were too few studies to meaningfully calculate heterogeneity (my recommendation) or recognise this in their limitations.

Line 388 – What is the total SUS score possible, can the authors report this as 76.5-88.1/Total possible as they have with TUQ score?

Lines 403-405 “In an era where the number of high-risk obstetric patients continues to rise, remote CTG monitoring may provide an avenue to alleviate the capacity strain on outpatient/inpatient services.” This is interpretation which belongs in the discussion section rather than the results section.

Line 455 I think vistations should be visits or attendances.

In the section on economic burden can the costs be presented in a single currency for comparison?

Discussion – Lines 583-585 I think represent the biggest challenge in this field. The diagnostic accuracy was comparable to conventional in hospital CTG. However, there is no evidence that conventional CTG improves perinatal outcome by reducing perinatal mortality. Surely, one of the key points here is that home CTG actually needs to provide some evidence of improved outcome before it can be introduced. Simply performing an ineffective test in another environment doesn’t mean that the technology should be adopted. I think the authors need to address this need to demonstrate improved outcome for women having antenatal CTG. I agree that there is a need for high-quality studies.

I think it is also worth considering the environmental impact of remote monitoring in the discussion section of the manuscript.

Reviewer #3: The authors are to be congratulated for a throrough sufficiently detailed Meta-Analysis with clearly presented Figures and Tables on a clinically important topic in antinatal care using adequate statistical analyses. The results support the conclusion of non-inferiority of remote cardiotocography to standard antenatal care. The manuscript should be accepted with wth minor revisions.

Minor issues.

Discussion:

The authors may consider to mention available information on risk stratification and its effect on their conclusion of ‚non-inferiority‘ of remote cardiotocography,e.g., the use of pregnancy risk scores to define high risk pregnancies that would likely profit from remote CTG monitoring as compared to standard of antinatal care (https://www.sciencedirect.com/science/article/abs/pii/S0301211524005074).

It is likely, that remote CTG monitoring would be superior to the standard of care in the high risk group of pregnancies if tools to predict poor outcome would be employed on a large scale.

Line 604: omit ‚a‘

Reviewer #4: This systematic review comprehensively examined the available literature on home CTG monitoring, including clinical effectiveness, feasibility, diagnostic accuracy, clinical utility, and economic burden. The methods used for the review are appropriate and robust, the language is clear, and the conclusions are supported by the data.

Minor comments:

1. In the final sentences of the introduction's second paragraph, the authors differentiate their work from an existing review by emphasising their focus on home CTG monitoring. To ensure consistency and clarity, could the authors consider aligning the title more directly with this focus, for instance, by using "Home CTG" or "Remote Home CTG" instead of the broader term "Remote CTG"?

2. Regarding the grey literature search, the authors mention screening reference lists. Could they please clarify if additional, systematic methods were employed to identify grey literature? Specifically, was any search performed in dedicated grey literature databases such as OpenGrey or clinical trial registries like ClinicalTrials.gov?

3. In the results section, subsection Randomised controlled trials and GRADE assessment, the authors stated "Despite all assessing remote monitoring in high-risk participants”, however, in Table S5, Wang 2019 (54), High/Low risk patients: NR, please clarify which one is correct?

**Do you want your identity to be public for this peer review?** For information about this choice, including consent withdrawal, please see our Privacy Policy

Reviewer #1: No

Reviewer #2: No

Reviewer #3: No

Reviewer #4: No

**Figure resubmission:**

**Reproducibility:** To enhance the reproducibility of your results, we recommend that authors of applicable studies deposit laboratory protocols in protocols.io, where a protocol can be assigned its own identifier (DOI) such that it can be cited independently in the future. Additionally, PLOS ONE offers an option to publish peer-reviewed clinical study protocols. Read more information on sharing protocols at https://plos.org/protocols?utm_medium=editorial-email&utm_source=authorletters&utm_campaign=protocols

---

## [Decision Letter · Decision Letter 1]

21 Dec 2025

Remote home cardiotocography: a systematic review and meta-analysis

PDIG-D-25-00790R1

Dear Dr Le Vance,

We are pleased to inform you that your manuscript 'Remote home cardiotocography: a systematic review and meta-analysis' has been provisionally accepted for publication in PLOS Digital Health.

Best regards,

Chenxi Yang, PhD

Hongxing Luo, MD, PhD

Leo Anthony Celi, PhD

PLOS Digital Health

**Additional Editor Comments (if provided):**

**Reviewer Comments (if any, and for reference):**

Reviewer's Responses to Questions

**Comments to the Author**

Reviewer #2: All comments have been addressed

Reviewer #3: All comments have been addressed

Reviewer #4: All comments have been addressed

publication criteria?

Reviewer #2: Yes

Reviewer #3: Yes

Reviewer #4: Yes

3. Has the statistical analysis been performed appropriately and rigorously?

Reviewer #2: Yes

Reviewer #3: Yes

Reviewer #4: Yes

4. Have the authors made all data underlying the findings in their manuscript fully available (please refer to the Data Availability Statement at the start of the manuscript PDF file)?

Reviewer #2: Yes

Reviewer #3: Yes

Reviewer #4: Yes

5. Is the manuscript presented in an intelligible fashion and written in standard English?

Reviewer #2: Yes

Reviewer #3: Yes

Reviewer #4: Yes

Reviewer #2: I am satisfied with the authors responses to all of my comments and think this manuscript is now suitable for publication.

Reviewer #3: This reviewer's comments have been addressed adequately.

Reviewer #4: (No Response)

**Do you want your identity to be public for this peer review?** For information about this choice, including consent withdrawal, please see our Privacy Policy

Reviewer #2: No

Reviewer #3: **Yes: ** Arne Jensen, MD

Reviewer #4: **Yes: ** Yanan Hu
